# Sae2/CtIP prevents R-loop accumulation in eukaryotic cells

Nodar Makharashvili[1,2], Sucheta Arora[1,2†], Yizhi Yin[1,2†], Qiong Fu[3], Xuemei Wen[2], Ji-Hoon Lee[2], Chung-Hsuan Kao[2], Justin WC Leung[4], Kyle M Miller[2], Tanya T Paull[1,2]*

[1]Howard Hughes Medical Institute, The University of Texas at Austin, Austin, United states; [2]Department of Molecular Biosciences, The University of Texas at Austin, Austin, United States; [3]Gastrointestinal Malignancy Section, Thoracic and Gastrointestinal Oncology Branch, Center for Cancer Research, National Cancer Institute, National Institutes of Health, Bethesda, United States; [4]Department of Radiation Oncology, University of Arkansas for Medical Sciences, Little Rock, United States

**Abstract** The Sae2/CtIP protein is required for efficient processing of DNA double-strand breaks that initiate homologous recombination in eukaryotic cells. Sae2/CtIP is also important for survival of single-stranded Top1-induced lesions and CtIP is known to associate directly with transcription-associated complexes in mammalian cells. Here we investigate the role of Sae2/CtIP at single-strand lesions in budding yeast and in human cells and find that depletion of Sae2/CtIP promotes the accumulation of stalled RNA polymerase and RNA-DNA hybrids at sites of highly expressed genes. Overexpression of the RNA-DNA helicase Senataxin suppresses DNA damage sensitivity and R-loop accumulation in Sae2/CtIP-deficient cells, and a catalytic mutant of CtIP fails to complement this sensitivity, indicating a role for CtIP nuclease activity in the repair process. Based on this evidence, we propose that R-loop processing by 5' flap endonucleases is a necessary step in the stabilization and removal of nascent R-loop initiating structures in eukaryotic cells.
DOI: https://doi.org/10.7554/eLife.42733.001

*For correspondence: tpaull@utexas.edu

†These authors contributed equally to this work

Competing interests: The authors declare that no competing interests exist.

## Introduction

Double-strand breaks in DNA are known to be lethal lesions in eukaryotic cells, and can be an important source of genomic instability during oncogenic transformation because of the possibility of mis-repair, translocations, and rearrangements that initiate from these lesions (*Aparicio et al., 2014*). The repair of double-strand breaks in eukaryotes occurs through pathways related to either non-homologous end joining or homologous recombination, although many variations on these basic pathways can occur in cells depending on the cell cycle phase, the extent of DNA end processing that occurs, which enzymes are utilized to do the processing, and what resolution outcomes predominate (*Ceccaldi et al., 2016*; *Symington, 2016*; *Symington and Gautier, 2011*).

The budding yeast enzyme Sae2 and its mammalian ortholog CtIP are important for DNA end processing in eukaryotes and has been shown to act in several ways to facilitate the removal of the 5' strand at DNA double-strand breaks (*Makharashvili and Paull, 2015*; *Symington, 2016*). Phosphorylated Sae2/CtIP promotes the activity of the Mre11 nuclease in the Mre11/Rad50/Xrs2(Nbs1) (MRX(N)) complex, which initiates the trimming of the DNA end on the 5' strand (*Cannavo and Cejka, 2014*). Sae2/CtIP and MRX(N) promote the removal of the Ku heterodimer, which acts as a block to resection during non-homologous end joining (*Cannavo and Cejka, 2014*; *Reginato et al., 2017*; *Wang et al., 2017*) and also recruit the long-range 5' to 3' nucleases Exo1 and Dna2 which

do extensive processing of the ends (*Cejka et al., 2010*; *Myler et al., 2016*; *Nicolette et al., 2010*; *Niu et al., 2010*; *Shim et al., 2010*).

In addition to the activities of Sae2/CtIP that promote MRX(N) functions, the protein has also been shown to possess intrinsic nuclease activity that is important for the processing of breaks, particularly those formed in the context of protein lesions, radiation-induced DNA damage, or campto-thecin (CPT) damage during S phase (*Symington, 2016*; *Chanut et al., 2016*; *Makharashvili et al., 2014*; *Wang et al., 2014*). Nuclease-deficient CtIP fails to complement human cells deficient in CtIP for survival of radiation-induced DNA damage while homologous recombination at restriction enzyme-induced break sites is comparable to wild-type-complemented cells (*Makharashvili et al., 2014*; *Wang et al., 2014*).

Eukaryotic cells lacking Sae2 or CtIP were also observed years ago to be hypersensitive to topo-isomerase one (Top1) poisons such as CPT (*Deng et al., 2005*; *Sartori et al., 2007*). Top1 bound to CPT is stalled in its catalytic cycle in a covalent tyrosine 3' linkage with DNA, creating a protein-linked DNA strand adjacent to a 5' nick (*Pommier, 2006*). This lesion targets one DNA strand but can lead to double-strand breaks during replication. Importantly, Top1 is highly active at sites of ongoing transcription due to the need for the release of topological stress in front of and behind the RNA polymerase (*Liu and Wang, 1987*; *Pommier et al., 2016*; *Zhang et al., 1988*). Considering that MRX(N) complexes as well as Sae2/CtIP are essential for the removal of 5' Spo11 conjugates during meiosis (*McKee and Kleckner, 1997*; *Neale et al., 2005*; *Prinz et al., 1997*) coincident with their role in 5' strand processing, it is surprising that Sae2/CtIP-deficient cells exhibit such sensitivity to 3' single-strand lesions, and the mechanistic role that Sae2/CtIP plays at these lesions is currently unknown.

In recent years, it has become clear that transcription can play a major role in promoting genomic instability by forming stalled transcription complexes and RNA-DNA hybrids in the genome (*Sollier and Cimprich, 2015*). Stable annealing of nascent RNA with the DNA template strand can occur at sites of stalled RNA polymerase complexes, and the 'R-loops' formed in this way can block replication as well as other DNA transactions (*Santos-Pereira and Aguilera, 2015*). A wealth of evidence accumulated recently suggests that these events can lead to single-strand and double-strand breaks in DNA that provide recombinogenic intermediates for misrepair events (*Costantino and Koshland, 2015*; *Hamperl et al., 2017*; *Huertas and Aguilera, 2003*).

The Sen1 protein in budding yeast has been shown to regulate many aspects of RNA biology, including termination of RNA polymerase II transcription, 3' end processing of mRNA, and dissociation of RNA-DNA hybrids (*Mischo et al., 2011*; *Steinmetz et al., 2006*; *Finkel et al., 2010*). The human ortholog of Sen1, Senataxin, has also been shown to resolve RNA-DNA hybrids (*Skourti-Stathaki et al., 2011*; *Yüce and West, 2013*; *Cohen et al., 2018*) as well as to associate with replication forks to protect fork integrity when traversing transcribed genes (*Alzu et al., 2012*). Mutations in the gene encoding Senataxin are responsible for the neurodegenerative disorder Ataxia with Oculomotor Apraxia two as well as an early-onset form of amyotrophic lateral sclerosis (*Groh et al., 2017*).

In this work we sought to understand the mechanistic basis of the hypersensitivity of Sae2/CtIP-deficient cells to Top1 poisons and other forms of DNA damage, finding unexpectedly that the sensitivity of these cells can be partially suppressed by overexpression of Sen1/Senataxin or RNaseH. Genetic evidence in yeast and human cells shows that Sae2/CtIP-deficient cells require Senataxin and RNaseH enzymes for survival of genotoxic agents and that these cells exhibit high levels of RNA polymerase stalling and R-loop formation at sites of active transcription, consistent with a failure to recognize or process RNA-DNA hybrids. Based on this evidence we propose that Sae2/CtIP has an important role in processing R-loops that promotes the action of RNA-DNA helicases and ultimately cell survival after DNA damage.

## Results

### Transcription termination factors rescue DNA damage sensitivity of sae2Δ and mre11 nuclease-deficient yeast cells

To test for an effect of transcriptional regulation on the *sae2Δ* phenotype in yeast, we overexpressed several different RNA Pol II-associated factors in the mutant strain. We found that overexpression of

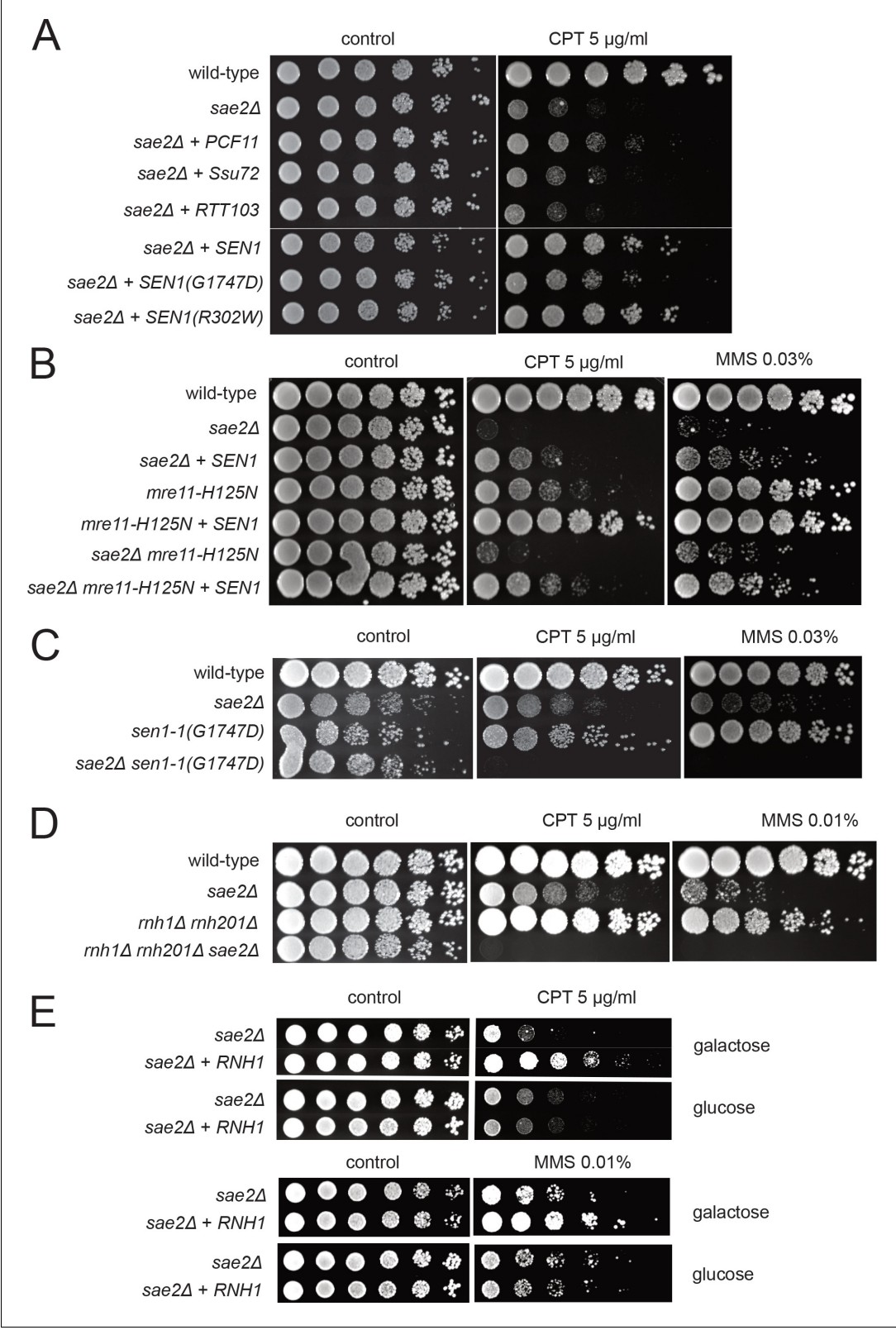

**Figure 1.** Transcription termination factors suppress DNA damage sensitivity of *sae2Δ* and *mre11* nuclease-deficient strains. (**A**) Full-length *PCF11*, *SSU72*, *RTT103*, *SEN1*, and *sen1* mutants G1747D and R302W were expressed from a 2μ plasmid in *sae2Δ* cells. Fivefold serial dilutions of cells expressing the indicated Sae2 alleles were plated on nonselective media (control) or media containing camptothecin (CPT, 5.0 μg/ml) and grown for 48 hr (control) or 70 hr (CPT). (**B**) *SEN1* was expressed from a 2μ plasmid in *sae2Δ*, *mre11-H125N*, and *sae2Δ mre11-H125N* cells and analyzed for CPT

*Figure 1 continued on next page*

Figure 1 continued

sensitivity as in (A). (C) Wild-type, sae2Δ, sen1-1(G1747D), and sae2Δ sen1-1(G1747D) strains were analyzed as in (A). (D) Wild-type, sae2Δ, Δrnh1 Δrnh201, and sae2Δ Δrnh1 Δrnh201 strains were analyzed as in (A). (E) sae2Δ strains with RNH1 expressed under the control of the GAL promoter were tested for sensitivity to CPT and MMS, on either galactose or glucose plates indicated.

DOI: https://doi.org/10.7554/eLife.42733.002

The following figure supplements are available for figure 1:

**Figure supplement 1.** Overexpression of *SSU72*, *NRD1*, *RTT103*, or *YSH1* does not complement Δ*sae2* strains for DNA damage sensitivity.

DOI: https://doi.org/10.7554/eLife.42733.003

**Figure supplement 2.** *SEN1* overexpression does not complement the resection deficiency in Δ*sae2* yeast strains.

DOI: https://doi.org/10.7554/eLife.42733.004

the termination factor Sen1 markedly improved survival of the *sae2Δ* strain to genotoxic agents (*Figure 1A*). *S. cerevisiae SEN1* encodes a helicase that is responsible for unwinding RNA-DNA hybrids and also promotes transcription termination through direct contact with RNA Pol II as well as 3′ end processing of RNA (*Porrua and Libri, 2015*). We also found that PCF11, a component of the cleavage and polyadenylation complex (CPAC) (*Grzechnik et al., 2015*; *Birse et al., 1998*), improves the survival of yeast strains lacking *SAE2* when tested for survival of CPT but there was little effect of overexpressing other proteins that also regulate transcription through RNA Pol II including *SSU72*, *RTT103*, *NRD1*, and *YSH1* (*Figure 1A* and *Figure 1—figure supplement 1*).

The ability of Sen1 overexpression to partially alleviate the toxicity of CPT was also observed with the Mre11 nuclease-deficient mutant *mre11-H125N* (*Moreau et al., 1999*) and particularly with the double mutant *sae2Δ mre11-H125N* (*Figure 1B*). A mutation located in the conserved helicase domain of Sen1 (G1747D) reduces the ability of Sen1 to overcome CPT toxicity in the *sae2Δ* strain (*Figure 1A*) but there was no effect of R302W, a mutation reported to block binding to the Rpb1 subunit of RNA Pol II (*Chinchilla et al., 2012*; *Finkel et al., 2010*). The *sen1-G1747D* mutant is deficient in transcription termination but not in 3′ end processing of RNA (*Mischo et al., 2011*), thus we conclude that the termination function of the Sen1 enzyme is important for the rescue of CPT sensitivity in *sae2Δ* strains. In contrast, Sen1 overexpression in *sae2Δ* cells has no effect on the efficiency of resection (*Figure 1—figure supplement 2*), as measured in an assay for single-strand annealing (*Vaze et al., 2002b*) previously shown be dependent on *SAE2* due to its importance in DNA end resection (*Clerici et al., 2005*).

To further investigate the genetic relationship between *SEN1* and *sae2Δ* phenotypes, we deleted the *SAE2* gene in a *sen1-1* (G1747D) background. A complete deletion of *SEN1* is lethal (*DeMarini et al., 1992*); however, the *sen1-1* allele has been used as a hypomorphic mutant and is deficient in transcription termination and removal of R-loops in vivo (*Mischo et al., 2011*). Although the *sen1-1* strain is not sensitive to the levels of DNA damaging agents used here, a combination with *sae2Δ* generates extreme sensitivity to both CPT and MMS (*Figure 1C*). Synthetic sensitivity of *sen1-1* with other DNA repair mutant strains has previously been shown for HU exposure (*Mischo et al., 2011*). Since the Sen1 helicase acts to remove R-loops from genomic loci, we also tested whether *sae2Δ* strains show synthetic sensitivity to CPT in combination with deletions of RNase H enzymes which remove ribonucleotides from DNA. Deletion of both RNase H1 and H2 in a Δ*rnh1* Δ*rnh201* strain generates modest DNA damage sensitivity as previously shown (*Lazzaro et al., 2012*; *Zimmer and Koshland, 2016*; *Arudchandran et al., 2000*); however, this is further exacerbated by a deletion of *SAE2* (*Figure 1D*). Conversely, overexpression of *RNH1* in a *sae2Δ* strain from a galactose-inducible promoter partially rescues the strain upon CPT or MMS exposure (*Figure 1E*). Taken together, these results suggest that loss of Sae2, either by itself or in combination with Mre11 nuclease activity, generates a form of DNA damage sensitivity that requires efficient removal of RNA or ribonucleotides from DNA.

RNA-DNA hybrids form at sites of RNA polymerase pausing and can generate collisions between the transcription machinery and the replication fork (*Santos-Pereira and Aguilera, 2015*). We postulated that the source of lethal damage in a *sae2Δ* strain may be related to transcription-replication conflicts, based on the *SEN1* and RNaseH observations above. To test this idea we synchronized wild-type and *sae2Δ* yeast strains with alpha factor and exposed the cells to CPT in either $G_1$ or S phases of the cell cycle. As expected, the *sae2Δ* strain showed marked sensitivity to CPT and this was specific to exposure in S phase (*Figure 2A*), suggesting that movement of replication forks

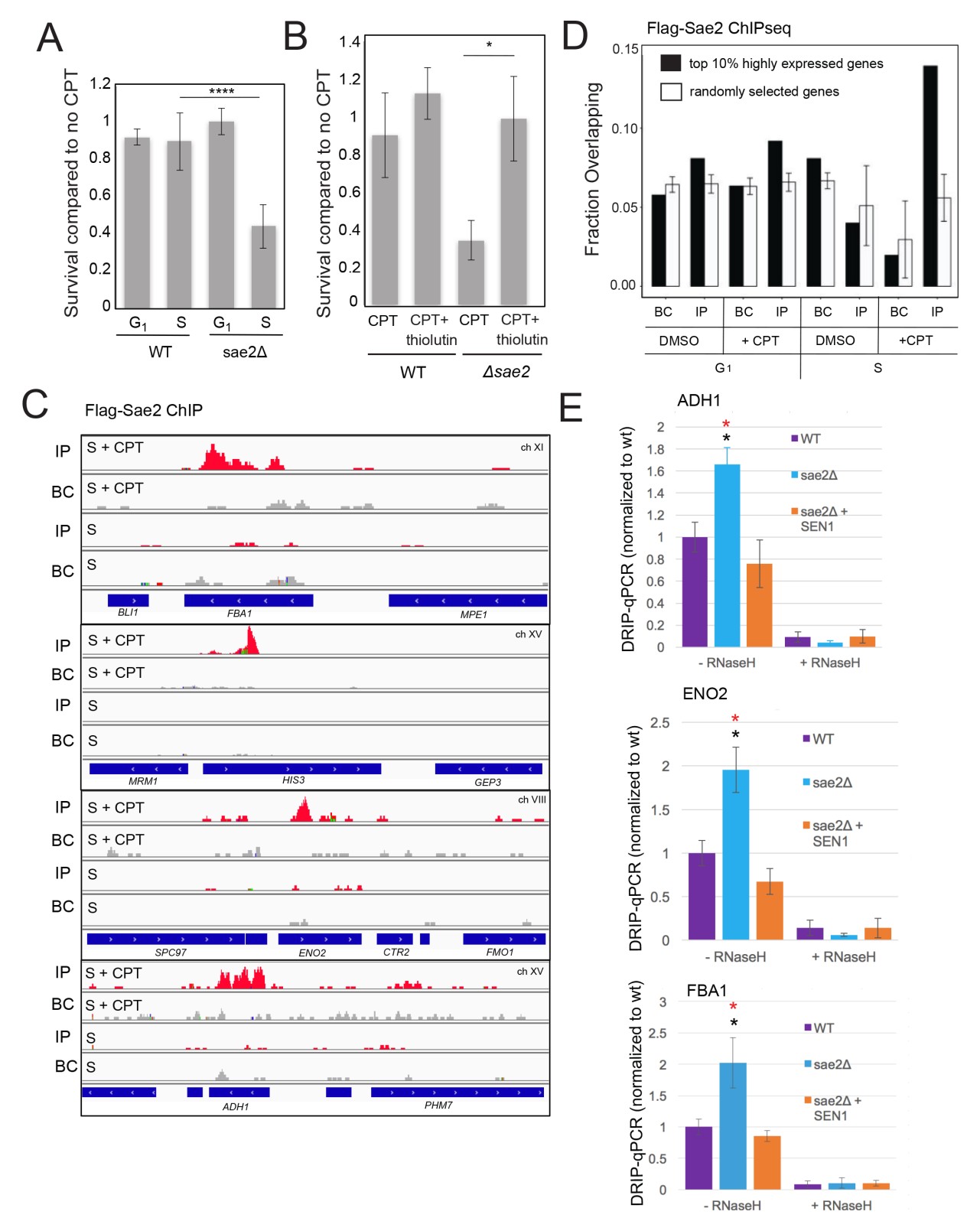

**Figure 2.** Sae2 associates with sites of high levels of transcription which accumulate R-loops in the absence of Sae2. (**A**) The survival of wild-type and *sae2Δ* strains was measured by exposing cells to camptothecin (100 µM for 2 hr) while in G$_1$ phase or S phase and plating cells on rich media. The percentage of viable colonies is shown relative to cells exposed to DMSO only, with three biological replicates (error bars represent standard deviation). (**B**) The survival of wild-type and *sae2Δ* strains was measured in the absence or presence of active transcription by exposing S phase cells to

*Figure 2 continued on next page*

*Figure 2 continued*

thiolutin (2.5 µg/ml for 30 min) or DMSO prior to camptothecin exposure in S phase as in (**A**). (**C**) Representative examples of Sae2-ChIP at the FBA1, HIS3, ENO2, and ADH1 genes in *sae2Δ* cells expressing Flag-Sae2, in S phase with CPT exposure as in (**A**). Reads from the immunoprecipitated sample are shown (IP) in comparison to control immunoprecipitations performed in the absence of Flag antibody (bead control, BC). (**D**) Data from Sae2 ChIP-seq was compared to previous data on transcription levels in wild-type yeast cells (*Nagalakshmi et al., 2008*; *Pelechano et al., 2010*; *Miura et al., 2008*)(See *Supplementary file 2*). The overlap between peaks identified by Sae2 ChIP-seq were compared to the top 10% of transcribed genes (486 genes; excluding rDNA loci) or a randomly chosen set of genes. The randomized set comparison was performed 1000 times. (**E**) R-loops were quantified at various loci using S9.6 immunoprecipitation in wild-type, *sae2Δ*, and *sae2Δ + SEN1* yeast strains, all with CPT treatment in S phase, as indicated. Levels of DNA sites enriched in S9.6 immunoprecipitations are shown relative to levels in wild-type cells using primers specific for ADH1 (ADH1-2), ENO2 (ENO2-3), and FBA1 (FBA1-2). Error bars represent standard deviation from four biological replicates. * indicates p < 0.05 comparing *sae2Δ* to wild-type (black asterisks) or comparing *sae2Δ* to *sae2Δ* plus *SEN1* (red asterisks) using 2-tailed Student's t-tests. S9.6 immunoprecipitations were also performed with RNaseH pretreatment of chromatin, '+RNaseH'.
DOI: https://doi.org/10.7554/eLife.42733.005

through Top1 DNA damage sites is important for the DNA damage sensitivity. We also tested for the effect of transcription on CPT survival by incubating cells with the general transcription inhibitor thiolutin (*Jimenez et al., 1973*). Inhibition of transcription completely alleviated the sensitivity of the *sae2Δ* strain to CPT, generating wild-type levels of survival (*Figure 2B*).

## Sae2 occupancy is elevated at sites of high transcription in S phase cells

To determine if the relationship between *SAE2* deletion and transcription is direct, we sought to determine the genomic locations of Sae2 protein binding in yeast. We performed ChIP assays using Flag-tagged Sae2 and analyzed the peaks in relation to the bead control (no antibody). This analysis revealed small peaks of Sae2 enrichment, primarily in the S phase +CPT sample (176 peaks identified by Model-based Analysis of ChIP-Seq v.2 (MACS2) (*Zhang et al., 2008*) after removal of overlaps with bead control) (*Figure 2C*, *Supplementary file 2*). In contrast, only 45 peaks were identified by this criteria in S phase in the absence of DNA damage. The locations of the sites enriched with CPT treatment did not correlate with sites of replication origins but were enriched for highly transcribed genes, measuring the overlap between the binding sites and the top 10% of highly transcribed genes (*Nagalakshmi et al., 2008*; *Pelechano et al., 2010*; *Miura et al., 2008*) in comparison to a randomly selected subset (*Figure 2D*). The enrichment for Sae2 occupancy at sites of high transcription was only observed with cells in S phase, not with the $G_1$ phase cells, and only in cells treated with CPT. Approximately 20% (9 of 45 Sae2 peaks in S phase and 36 of 176 Sae2 peaks in S phase with CPT) overlap with the sites of RNA-DNA hybrids measured in wild-type yeast cells in a previous study (*Wahba et al., 2016*).

## sae2Δ cells accumulate R-loops and stalled RNA pol II during CPT exposure

The results from these experiments suggested an involvement of transcription in the DNA damage sensitivity of *sae2Δ* cells, possibly related to an accumulation of R-loops. To test this hypothesis we used the S9.6 antibody to detect RNA-DNA hybrids (*Boguslawski et al., 1986*) in chromatin immunoprecipitation experiments comparing wild-type, *sae2Δ*, and *sae2Δ* with Sen1 overexpression. We synchronized yeast cells in S phase, exposed the cultures to CPT, and observed approximately 2-fold higher levels of R-loops in *sae2Δ* cells at the ADH1, ENO2, and FBA1 loci compared to the wild-type strain (*Figure 2E*). This signal was reduced with Sen1 expression and was sensitive to RNaseH treatment in vitro, consistent with this interpretation that R-loops accumulate at these loci in the absence of Sae2.

If Sae2 is localized to sites of high transcription and its loss is partially alleviated by enzymes that promote transcription termination, we reasoned that levels of RNA polymerase may be stalled at these sites in *sae2Δ* strains. To test this idea we utilized a tagged RNA Pol II strain (HTB-Rpb2) (*Schaughency et al., 2014*) to monitor the occupancy of RNA Pol II at sites in the genome with high levels of constitutive transcription where Sae2 was observed by ChIP-seq in *Figure 2C*. HTB-tagged Pol II complexes were isolated from wild-type, *sae2Δ*, and *sae2Δ* with *Sen1* overexpression strains that were synchronized in $G_1$ with alpha-factor and released into S phase. All of the strains showed very similar levels of RNA Pol II occupancy on the ADH1, ENO2, and FBA1 genes in the absence of

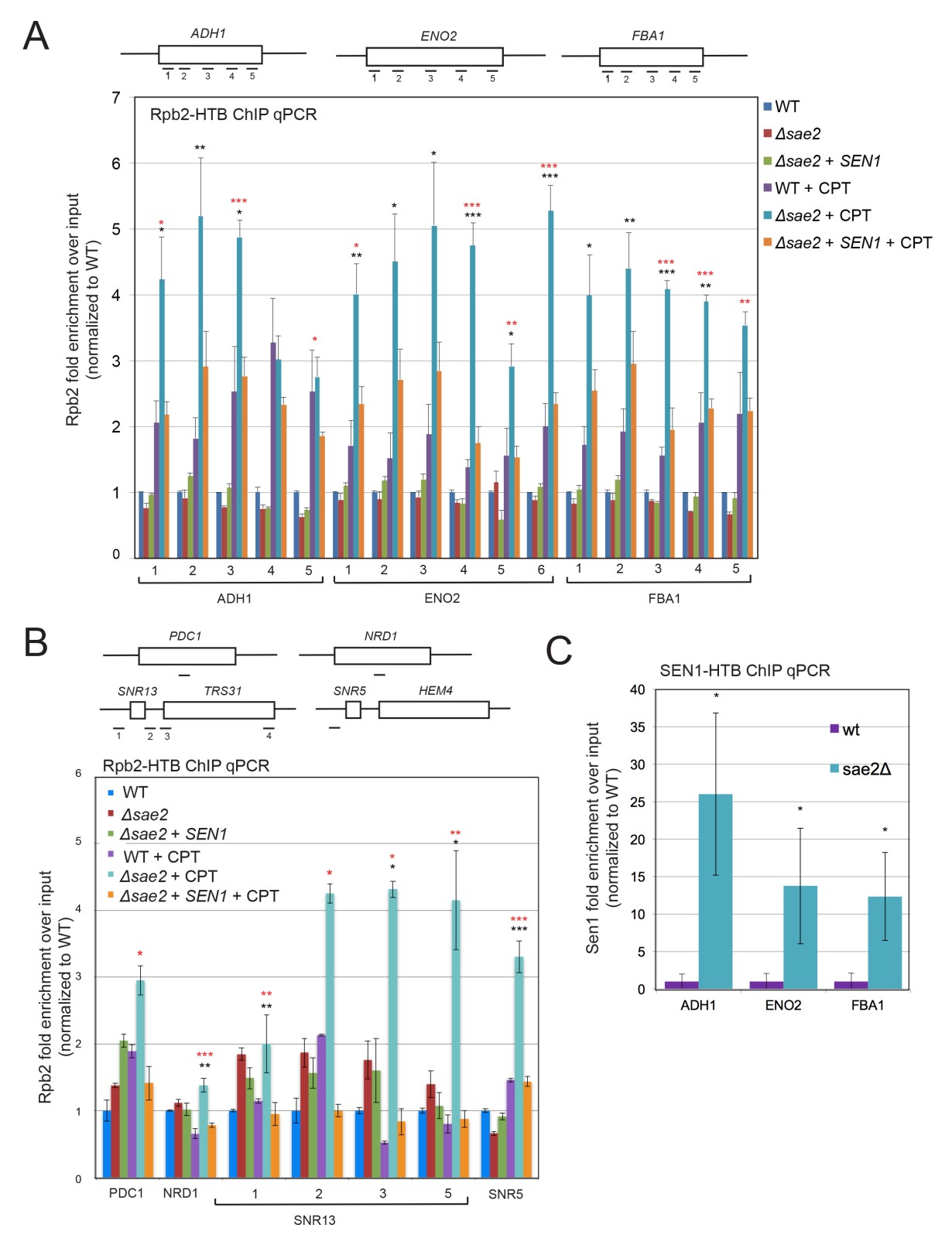

**Figure 3.** RNA Polymerase stalling at highly transcribed genes is exacerbated in *sae2Δ* strains with DNA damage. (**A**) HTB-tagged Rpb2 (a component of RNA Pol II) levels were measured at various genes during S phase in wild-type and *sae2Δ* strains with or without CPT exposure and in the presence or absence of overexpressed Sen1, as indicated. Enrichment relative to input DNA is shown, with all values normalized to values obtained with the wild-type strain in the absence of damage. Error bars represent standard error from six immunoprecipitations, two biological samples with three technical

*Figure 3 continued on next page*

*Figure 3 continued*

replicates of the IP per sample. Approximate locations of primer sets relative to the gene are shown. * indicates p < 0.05, **p < .005, ***p < 0.0005, comparing *sae2Δ* to wild-type (black asterisks) or comparing *sae2Δ* to *sae2Δ* plus *SEN1* (red asterisks) using 2-tailed Student's t-tests. (B) HTB-tagged Rpb2 levels were measured at various genes as in (A). *SNR5* and *SNR13* are non-coding nucleolar RNAs. (C) HTB-tagged Sen1 levels were measured at various genes as in (A) with CPT treatment. Primer sets used were ADH1-2, ENO2-4, and FBA1-3.
DOI: https://doi.org/10.7554/eLife.42733.006

DNA damage (*Figure 3A*). In contrast, when the strains were released into S phase in the presence of CPT, the wild-type strain showed an average of 1.5 to 2-fold higher levels of RNA Pol II occupancy, while *sae2Δ* strains exhibited 2.5 to 5.5-fold higher levels of polymerase stalling (*Figure 3A*). Importantly, *sae2Δ* with Sen1 overexpression showed reduced levels of polymerase occupancy, similar to the wild-type strain with CPT.

We also examined RNA Pol II occupancy at other genomic locations where transcription termination has previously been shown to be regulated by Sen1. We do observe CPT-induced RNA Pol II stalling at the *PDC1* or *NRD1* genes, both known to be dependent on Sen1 (*Alzu et al., 2012*; *Grzechnik et al., 2015*), albeit at a lower level than the genes identified in our Sae2 ChIP experiment. Obvious pausing of RNA Pol II in the absence of Sae2 was also induced by CPT at the snoRNA genes *SNR13* and *SNR5* and the associated gene *TRS31* (*Figure 3B*), which have been shown to be transcribed by RNA Pol II and exhibit termination read-through, altered RNA Pol II occupancy, and R-loops in the absence of *Sen1* function (*Steinmetz et al., 2006*; *Grzechnik et al., 2015*). Overall, these results are consistent with the hypothesis that Sae2 is present at a subset of highly transcribed genes during CPT exposure, and that high levels of toxic R-loops form at these sites in *sae2Δ* strains which can be reduced by *Sen1*.

Based on these results we considered the possibility that Sen1 is dependent on Sae2 for recruitment to sites of stalled transcription. To examine this we used an HTB-tagged Sen1 strain (*Creamer et al., 2011*) and monitored Sen1 recruitment during CPT treatment at a subset of the genomic locations where we observed Sae2 occupancy and RNA polymerase accumulation. This showed that Sen1 is present at these sites (ADH1, ENO2, FBA1) in the absence of *SAE2* (*Figure 3C*), thus the binding of the helicase to these sites is Sae2-independent.

## The CPT sensitivity of CtIP deficient cells is rescued by over-expression of senataxin or inhibition of transcription

In mammalian cells, the Sae2 ortholog CtIP (*Sartori et al., 2007*) promotes resection of DNA double-strand breaks in conjunction with the MRN complex in mammalian cells (*Makharashvili and Paull, 2015*). It was previously shown that depletion of CtIP generates extreme sensitivity to topoisomerase poison induced DNA lesions, particularly CPT-induced DNA damage (*Sartori et al., 2007*; *Huertas and Jackson, 2009*; *Nakamura et al., 2010*; *Makharashvili and Paull, 2015*). To examine the role of CtIP in human cells we used a U2OS cell line with a stably integrated doxycycline-inducible CtIP shRNA cassette. The cells were complemented with either shRNA-resistant wild-type eGFP-CtIP, or with vector only (*Figure 4—figure supplement 1*). As expected, depletion of CtIP greatly diminishes cell survival in the presence of CPT, and re-expression of wild-type CtIP rescues the CPT sensitivity (*Figure 4A*).

Based on the yeast experiments with *SAE2*, we hypothesized that overexpression of a helicase that specifically resolves RNA-DNA hybrids should rescue the sensitivity of CtIP-depleted cells to CPT exposure. To test this idea, we complemented the CtIP-depleted cells with the C-terminal domains (a.a. 1851–2677) of human Senataxin, the ortholog of yeast Sen1 that also acts to remove RNA-DNA hybrids and promotes transcription termination (*Porrua and Libri, 2015*). Remarkably, overexpression of Senataxin in the CtIP-depleted cells completely rescues the CPT sensitivity; however, unlike the yeast experiments, no effect of human PCF11 expression was detected (*Figure 4A*). Although Senataxin expression rescued the survival of CtIP-depleted cells after CPT exposure, only partial suppression of the defects in growth and DNA end resection were observed (*Figure 4—figure supplement 1*). These results also suggest that inhibition of transcription could rescue the CPT sensitivity of CtIP-depleted cells. Indeed, we observe that pre-treatment of the cells with DRB, an inhibitor of RNA polymerase II-dependent RNA synthesis, partially rescues the CPT sensitivity caused

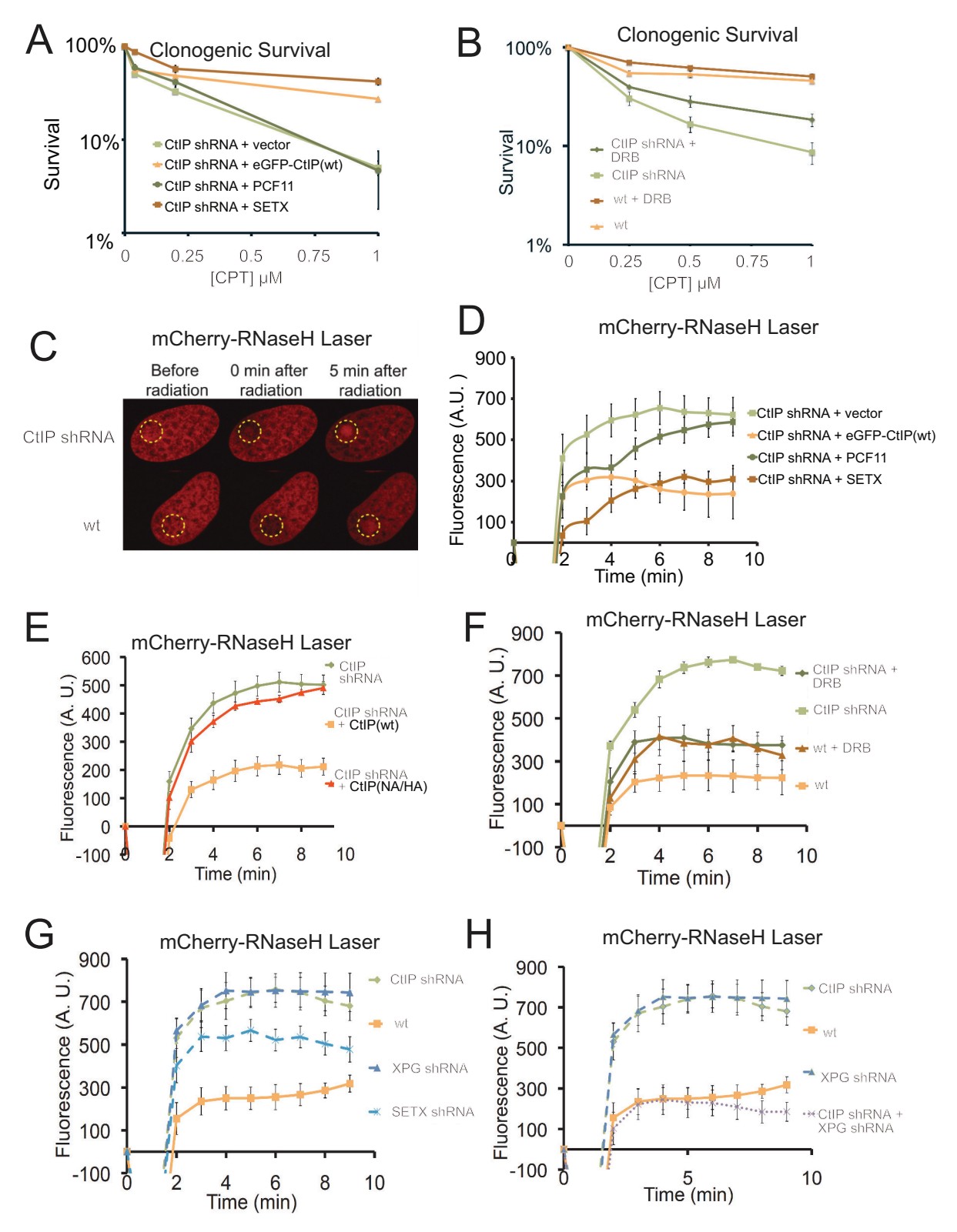

**Figure 4.** CtIP deficiency induces R-loop accumulation at sites of DNA damage. (**A**) CtIP-depleted U2OS cells complemented with vector only or constructs overexpressing eGFP-CtIP, PCF11, or Senataxin (C-terminus) as indicated were exposed to increasing concentrations of CPT (1 hr). Cell viability was determined by clonogenic survival assay in comparison to untreated cells. Results are shown from three biological replicates and error bars represent S. D. (**B**) Wild-type or CtIP-depleted U2OS cells were pre-treated with DRB (20 µM) prior to CPT exposure and cell viability was determined as

*Figure 4 continued on next page*

*Figure 4 continued*

in (A). (C) Live cell imaging was performed with U2OS cells stably expressing RNaseH$^{D10R-E48R}$-mCherry. The circle indicates the site of laser damage. (D) CtIP-depleted U2OS cells were complemented with eGFP-CtIP, PCF11, or Senataxin (C-terminus) as indicated and RNaseH$^{D10R-E48R}$-mCherry accumulation at the laser damage was quantified. The average of 4 cells is shown and error bars represent S. E. M. (E) RNaseH$^{D10R-E48R}$-mCherry accumulation at laser damage sites was measured in CtIP-depleted U2OS cells complemented with wild-type or nuclease deficient ('NA/HA') eGFP-CtIP as in (D); n > 12, error bars represent S.E.M. (F) Wild-type or CtIP-depleted U2OS cells were pre-treated with DRB (20 μM) prior to measurement of RNaseH$^{D10R-E48R}$-mCherry accumulation at laser damage sites as in (D); n > 5; error bars represent S.E.M. (G) RNaseH$^{D10R-E48R}$-mCherry accumulation at laser damage sites was measured in wild-type or CtIP-depleted, XPG-depleted, or Senataxin (SETX)-depleted U2OS cells as in (D); n = 5; error bars represent S.E.M. (H) RNaseH$^{D10R-E48R}$-mCherry accumulation at laser damage sites was measured in wild-type, CtIP-depleted, XPG-depleted, or both CtIP/XPG-depleted U2OS cells as in (D); n > 8, error bars represent S.E.M. Results shown are representative of several experiments performed.

DOI: https://doi.org/10.7554/eLife.42733.007

The following figure supplements are available for figure 4:

**Figure supplement 1.** Senataxin overexpression partially rescues growth but not resection in CtIP-depleted human cells.

DOI: https://doi.org/10.7554/eLife.42733.008

**Figure supplement 2.** There is no effect of CtIP depletion on mCherry independent of RNaseH.

DOI: https://doi.org/10.7554/eLife.42733.009

by CtIP deficiency (*Figure 4B*). Thus, similar to yeast, reduction of transcription during DNA damage exposure alleviates the effects of CtIP deficiency in human cells, suggesting a conserved mechanism for the repair of Top1 lesions associated with aberrant transcription.

## CtIP deficiency induces accumulation of R-loops at laser micro-irradiation sites

Similar to topoisomerase-DNA adducts, UVA laser-induced DNA crosslinks present a physical barrier to RNA polymerase that could stall transcription and promote the formation of complex DNA lesions including R-loops. We utilized a live cell assay with lentiviral expression of a bacterial RNaseH catalytic mutant fused to mCherry (RNaseH$^{D10R-E48R}$-mCherry) that acts as a sensor for R-loops in the genome (*Bhatia et al., 2014a*). The RNaseH$^{D10R-E48R}$-mCherry sensor was expressed in the wild-type or CtIP-depleted U2OS cells described above, which were laser micro-irradiated in a small area of the nucleus. Measurement of the accumulation of RNaseH$^{D10R-E48R}$-mCherry signal over time in these cells showed that the recruitment of RNaseH occurs to a higher intensity in CtIP-depleted cells in comparison to cells complemented with wild-type CtIP, suggesting that higher levels of R-loops are formed in the absence of CtIP (*Figure 4C,D*). A similar control experiment examining mCherry alone did not show this pattern, confirming the effect is specific to the RNaseH fusion (*Figure 4—figure supplement 2*). Since we found that overexpression of Senataxin rescues the CPT sensitivity of CtIP-depleted cells, we reasoned that it might also rescue the higher levels of R-loop accumulation in the absence of CtIP. We expressed the C-terminal domains of human Senataxin in the CtIP-depleted cells, and measured the levels of laser-induced R-loops by the live cell imaging method. Here also we found that with Senataxin expression, the recruitment of RNaseH in the CtIP-depleted cells is reduced to the levels observed in cells expressing wild-type CtIP (*Figure 4D*). Similar to the CPT survival assay results, we found that the PCF11 was not able to rescue CtIP deficiency for reduction of R-loop levels after laser micro-irradiation.

Previous work on CtIP in vitro identified mutants that exhibit lower levels of endonuclease activity on flap structures in comparison to the wild-type protein (*Makharashvili et al., 2014*; *Wang et al., 2014*). Here we used the N289A/H290A (NA/HA) mutant allele also containing shRNA-resistant mutations to complement CtIP-depleted cells and found that this mutant failed to reduce the increased RNaseH recruitment to damage sites that occurs in CtIP-deficient cells (*Figure 4E*, *Figure 4—figure supplement 1*). Thus the nuclease activity of CtIP appears to play a role in DNA damage recognition and/or processing that helps to prevent R-loop accumulation in human cells.

## The accumulation of R-loops in CtIP deficient cells is dependent on transcription

To further test the role of active transcription in R-loop accumulation, we pre-treated cells with 5,6-dichloro-1-beta-D-ribofuranosylbenzimidazole (DRB), a RNA polymerase II inhibitor, and found that inhibition of transcription rescues the R-loop accumulation phenotype caused by CtIP deficiency

(*Figure 4F*). As overexpression of Senataxin reduces R-loop formation in CtIP-depleted cells after DNA damage, we expected that depletion of this enzyme would have the opposite effect. Indeed, U2OS cells depleted of Senataxin also exhibit increased R-loop formation after laser-induced DNA damage (*Figure 4G*, *Figure 4—figure supplement 1*), consistent with previous observations of DNA damage sensitivity in Senataxin mutant cells (*Suraweera et al., 2007*; *Lavin et al., 2013*).

The XPG protein, a component of nucleotide excision repair (*Fagbemi et al., 2011*), has been shown to be important in the resolution of R-loops (*Sollier et al., 2014*). As a comparison to CtIP, we also assessed whether the deficiency of this protein would also lead to high R-loop levels after DNA damage. As expected, we observed that the XPG deficient cells accumulate more R-loops after laser induced DNA breaks in comparison to wild-type cells (*Figure 4G*, S2G). Interestingly, however, concurrent depletion of both CtIP and XPG showed R-loops at levels comparable to wild-type untreated cells (*Figure 4H*). This result suggests that R-loops do not form efficiently in the absence of both CtIP and XPG, or that they are not recognized efficiently by the RNaseH$^{D10R-E48R}$-mCherry protein in the absence of both CtIP and XPG.

## R-loop accumulation in CtIP-depleted cells without exogenous damage

Considering that deletion of the gene encoding CtIP is cell-lethal even in the absence of exogenous damage (*Polato et al., 2014*; *Chen et al., 2005*), we also hypothesized that R-loops might accumulate in CtIP-depleted cells under normal growth conditions. To address this question, we again used the RNaseH$^{D10R-E48R}$-mCherry sensor but in this case monitored its accumulation in undamaged cells by fluorescence activated cell sorting (FACS), using a modified technique reported previously with a fragment of RNaseH fused to GFP (*Bhatia et al., 2014a*). Using this procedure, unbound RNaseH$^{D10R-E48R}$-mCherry protein is removed from the nucleoplasm by detergent extraction, while protein bound to chromatin is retained. This analysis showed a statistically significant increase in RNaseH$^{D10R-E48R}$-mCherry fluorescence intensity in CtIP-depleted U2OS cells in all cell cycle phases (*Figure 5A*). Analysis of mCherry alone in these cells showed no differences with CtIP depletion (*Figure 4—figure supplement 2*). This was also observed in XPG-depleted cells, consistent with the observed effects of both XPG and CtIP with laser damage as shown in *Figure 4*. Similarly, we examined R-loop accumulation in CtIP-depleted cells complemented with the nuclease-deficient form of CtIP (N289A/H290A, NA/HA, *Figure 4—figure supplement 2*) and found the levels of RNaseH$^{D10R-E48R}$-mCherry fluorescence intensity identical to that of CtIP-depleted cells (*Figure 5B*), suggesting that the nuclease activity of CtIP is necessary for R-loop resolution even in cells that are not exposed to exogenous damage.

To confirm these results with a different technique, we also utilized the S9.6 antibody, which recognizes RNA-DNA hybrids and has been widely used as a probe for R-loops in cells (*Santos-Pereira and Aguilera, 2015*; *Boguslawski et al., 1986*; *García-Rubio et al., 2018*). We used immunofluorescence signal from S9.6 antibody in control as well as CtIP-depleted, XPG-depleted, and Senataxin-depleted U2OS cells and quantified the level of signal per cell. As this antibody also recognizes the components of nucleoli, the S9.6 signal that colocalized with these organelles was subtracted from the total signal, and the result was normalized by the area of the nucleus. We found that undamaged CtIP-depleted or XPG-depleted cells have significantly more R-loops than their wild-type counterparts (*Figure 5C,D*). These findings suggest that CtIP is responsible for the prevention and/or resolution of R-loops, in normally growing cells as well as in cells exposed to DNA damage. We also observed that concurrent depletion of CtIP and XPG resulted in a lower level of R-loop accumulation (*Figure 5D*), similar to the result with laser-induced damage (*Figure 4H*). This observation of lower RNA-DNA hybrids in the absence of both nucleases is thus not specific to laser damage or to the sensor used for R-loop detection.

## Global patterns of transcription are altered with CtIP depletion

CtIP was originally identified as a binding partner of C-terminal Binding Protein (CtBP), a transcriptional co-regulator (*Schaeper et al., 1998*), and also binds directly to Rb and the tumor suppressor BRCA1 which also has been widely reported to affect transcription and R-loop formation (*Monteiro et al., 1996*; *Scully et al., 1997*; *Takaoka and Miki, 2018*; *Yu et al., 1998*; *Hatchi et al., 2015*). Considering our results with CtIP and R-loop accumulation, we asked whether depletion of CtIP has global effects on transcription patterns by analyzing mRNA using RNA-seq. We performed

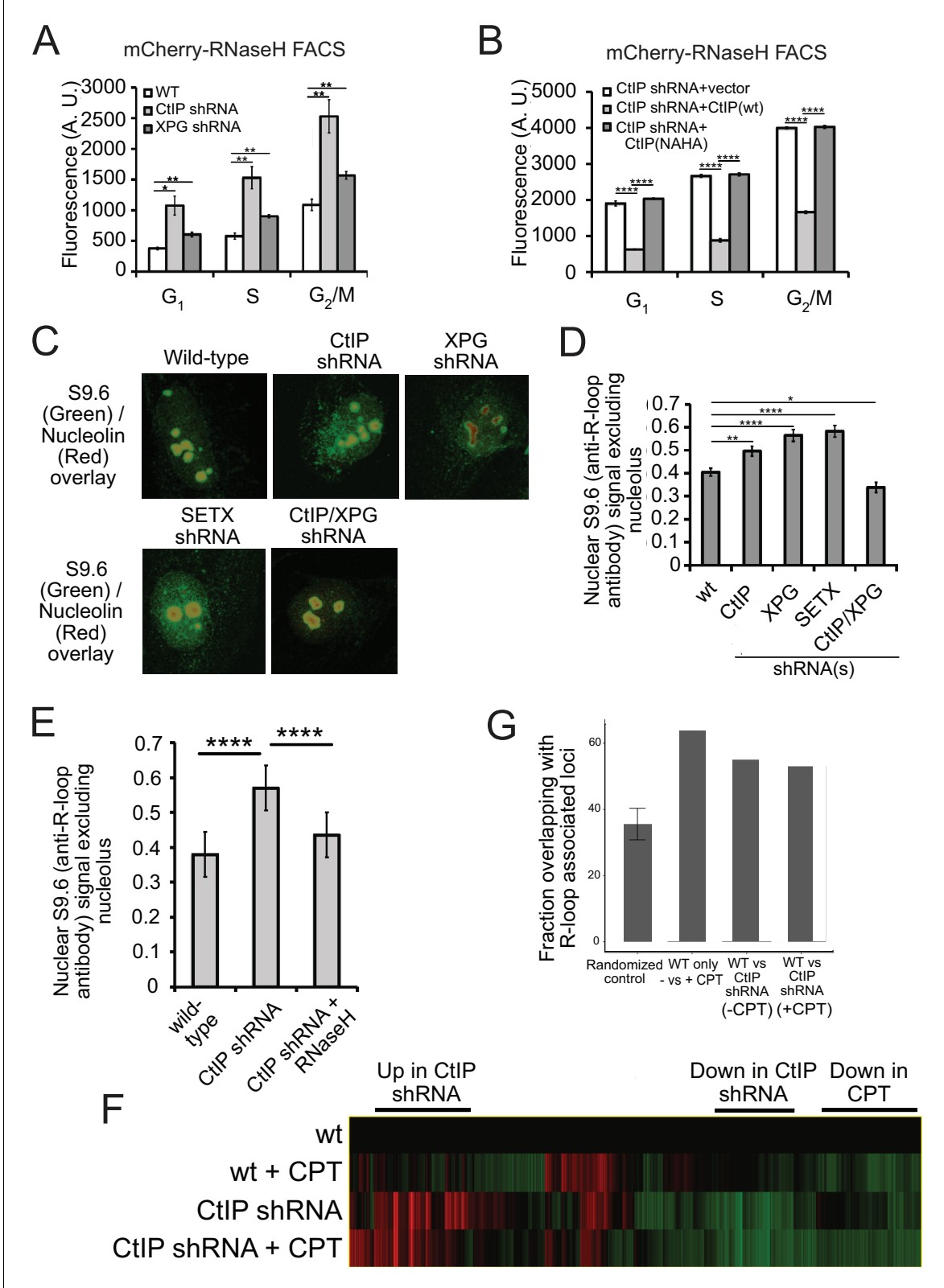

**Figure 5.** CtIP depletion affects R-loop accumulation and transcription in human cells. (**A**) DNA-RNA hybrids were quantified in undamaged U2OS cells by monitoring chromatin-bound RNaseH$^{D10R-E48R}$-mCherry by FACS. 10,000 cells were counted in each of 3 biological replicates; error bars represent S. D. * and ** denote $p < 0.05$ or 0.01, respectively in Student's two-tailed T test with comparisons as indicated. (**B**) RNA/DNA hybrids were quantified in undamaged, CtIP-depleted U2OS cells complemented with either vector only, eGFP-CtIP(wt), or nuclease-deficient eGFP-CtIP(NA/HA) as in (**A**). 10,000

*Figure 5 continued on next page*

*Figure 5 continued*

cells were counted in each of 3 biological replicates; error bars represent S. D. **** denotes p < 0.0001 using Student's two-tailed T test with comparisons as indicated. (C) S9.6 antibody was used to monitor RNA-DNA hybrids in wild-type or CtIP-depleted U2OS cells. Anti-Nucleolin was used as a marker for the nucleolus. (D) Quantification of S9.6 antibody signal in wild-type, CtIP-depleted, XPG-depleted, Senataxin-depleted, or double CtIP/ XPG-depleted U2OS cells as indicated. Signal overlapping the nucleolin signal was excluded from the analysis; n > 50, Error bars represent S.E.M. *, **, and **** denote p < 0.05, 0.01, and 0.0001, respectively, using Student's two-tailed T test with comparisons as indicated. (E) Quantification of S9.6 antibody signal in wild-type, CtIP-depleted, or CtIP-depleted plus RNaseH overexpression in U2OS cells as in (D). (F) Wild-type or CtIP-depleted U2OS cells were exposed to CPT (5 µM) or were untreated before harvesting of cellular mRNA. Analysis of transcripts by RNA-seq and hierarchical clustering of transcripts from 21,412 genes is shown as a heat map (red for over-expressed, black for unchanged expression, and green for under-expressed genes) in comparison to wild-type undamaged cells (see *Supplementary file 3*). (G) Statistical comparisons of overlap between the top 100 differentially expressed genes as ranked by DESeq differential expression p-value and DRIPc-seq peaks from GEO dataset GSE70189. Randomly picked genes were compared to this dataset ('randomized control') with the average of 35.56% and standard deviation 4.76% (estimated from 1000 simulations). Genes with significant differences between wild-type and CtIP-depleted cells were identified in the absence of DNA damage ('WT vs CtIP shRNA (-CPT')) as well as with CPT treatment ('WT vs CtIP shRNA (+CPT')) and were compared with the DRIPc-seq dataset. All three values are above the 99% confidence intervals for the null hypothesis that the genes showing the most evidence of differential expression overlapped the peak regions at the same rate as randomly selected genes.

DOI: https://doi.org/10.7554/eLife.42733.010

The following figure supplement is available for figure 5:

**Figure supplement 1.** CtIP-depleted human cells show alterations in transcription of endogenous genes.

DOI: https://doi.org/10.7554/eLife.42733.011

this analysis on U2OS cells expressing control or CtIP-specific shRNA and examined both CPT-treated and untreated conditions. RNA levels were quantified for 30,769 transcripts. CtIP depletion was found to alter 5013 (~16%) of these transcripts, with both increases (2,578) and decreases (2,435) observed relative to the control cells. Unsupervised hierarchical clustering was used to analyze the transcripts (*Figure 5E*) (*Supplementary file 3*). We found that there are transcriptional changes associated with CPT exposure in U2OS cells: 1285 transcripts were upregulated upon CPT treatment, and 1158 transcripts were downregulated. Interestingly, a comparison of the genes affected by CPT exposure to a previous dataset of genes showing R-loop accumulation indicates an overlap significantly higher than would be predicted by chance (*Figure 5F*). Over 60% of the genes affected by CPT (either up or down) overlap with R-loop prone regions of the genome (*Ginno et al., 2012*) (see 'WT only, - vs +CPT', *Figure 5F*), whereas less than 40% overlap with DNA-RNA immunoprecipitation (DRIP) positive genes occurs with a randomized control. A similar analysis of the gene set identified in CtIP-depleted cells compared to normal cells also indicates a higher than expected overlap (~50%), suggesting that depletion of CtIP has effects on transcription that are correlated with R-loop-prone regions of the genome.

## Quantitation of RNA-DNA hybrids in CtIP-depleted cells

To investigate the role of CtIP in R-loop accumulation at specific loci, we first generated an inducible genomic cassette containing a mouse class switch region (Sγ3) that has previously been shown to accumulate R-loops (*Huang et al., 2006*). We performed DRIP with the S9.6 antibody followed by quantitative PCR for this locus in U2OS cells and found that the levels of R-loops increase with doxycycline-induced transcription, even in wild-type cells (*Figure 6A*). The DRIP signal was removed by RNaseH treatment, confirming the specificity of the S9.6 antibody. A much larger increase in R-loops was found in CtIP-depleted cells, however, while the overall levels of transcripts are similar in both cases (*Figure 6A*, *Figure 6—figure supplement 1*).

We then examined several endogenous loci that have previously been reported to be prone to R-loop formation (*Bhatia et al., 2014b*; *Hatchi et al., 2015*). The β-actin gene has been reported to accumulate R-loops, particularly at the G-rich 'pause' sequences downstream of the coding region (*Hatchi et al., 2015*; *Skourti-Stathaki et al., 2014*). We observed higher levels of RNA-DNA hybrids at these regions in CtIP-depleted cells in the absence of exogenous DNA damage, which could be reduced to wild-type levels by overexpression of wild-type RNaseH (*Figure 6B*). We also tested XPG-depleted cells and observed an even higher level of R-loops under these conditions (*Figure 6C*). Similar results were observed on the UBF and CD30 genes (*Figure 6—figure supplements 2* and *3*), which we tested because levels of transcription are significantly lower in CtIP-

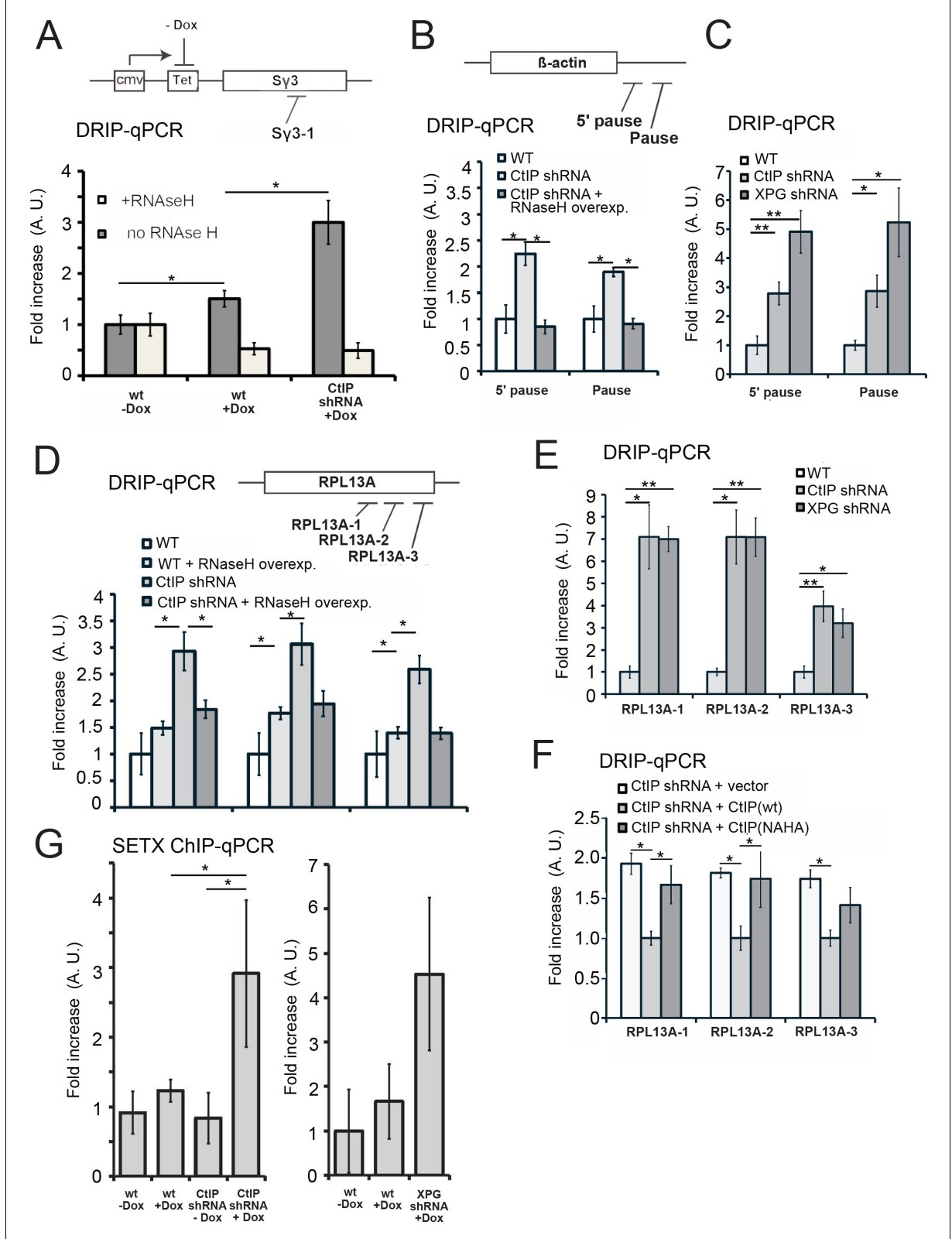

**Figure 6.** CtIP depletion induces accumulation of R-loops in human cells. (**A**) RNA/DNA hybrids were quantified by DRIP-qPCR in U2OS cells containing a stably integrated, doxycycline-inducible transgene containing murine Sγ3 repeats. R-loop accumulation was measured by immunoprecipitation with the S9.6 antibody and qPCR for the Sγ3 region in the absence or presence of transcription (-/+Dox) in wild-type or CtIP-depleted cells; n > 6, error bars represent S.E.M. (**B**) Quantification of RNA/DNA hybrids in U2OS cells at endogenous loci using DRIP-qPCR. Levels of hybrids were measured at the ß-

*Figure 6 continued on next page*

*Figure 6 continued*

actin gene, with CtIP or XPG depletion, and with RNaseH overexpression in cells, as indicated; n > 3, error bars represent S. D. (C), (D), (E). DRIP-qPCR of sites in the RPL13A gene are shown with CtIP or XPG depletion and RNaseH overexpression in cells as indicated. In (F), CtIP-depleted cells were complemented with wild-type eGFP-CtIP or nuclease-deficient NA/HA mutant. (G) Senataxin ChIP was performed in cells containing the doxycycline-inducible transgene containing murine Sγ3 repeats. Levels of SETX occupancy were quantified in the presence or absence of doxycycline and with CtIP depletion as indicated and are represented as fold changes relative to wild-type cells in the absence of dox. n = 3, error bars represent standard deviation. * and ** denote p < 0.05 or 0.01, respectively, using Student's two-tailed T test with comparisons as indicated.

DOI: https://doi.org/10.7554/eLife.42733.012

The following figure supplements are available for figure 6:

**Figure supplement 1.** Sγ3 transcript levels are similar in wild-type and CtIP-depleted cells.

DOI: https://doi.org/10.7554/eLife.42733.013

**Figure supplement 2.** CtIP depletion generates R-loops at the CD30 and UBF loci in human cells.

DOI: https://doi.org/10.7554/eLife.42733.014

**Figure supplement 3.** DRIP-qPCR signal is eliminated by RNaseH treatment in vitro.

DOI: https://doi.org/10.7554/eLife.42733.015

depleted cells (*Supplementary file 3*, *Figure 5—figure supplement 1*). We also examined RPL13A, a gene that has been identified as a region prone to R-loop formation (*Bhatia et al., 2014b*; *García-Rubio et al., 2018*). We tested three locations throughout the body of the RPL13A gene and found significantly higher levels of R-loops in CtIP-depleted cells which were reduced by overexpression of RNaseH in cells (*Figure 6D*) or by treatment of genomic DNA with RNaseH in vitro prior to the immunoprecipitation (*Figure 6—figure supplement 3*). We also observed high levels of R-loops at this gene in XPG-depleted cells compared to wild-type (*Figure 6E*). Lastly, the nuclease-deficient NA/HA allele of CtIP was expressed in CtIP-depleted cells, and R-loops were found to be approximately 1.5 to 2-fold higher in these cells in comparison to cells expressing the wild-type allele, similar to our observations in uncomplemented cells (*Figure 6F*).

Senataxin has been shown to localize at sites of transcription-replication conflicts and to travel with replication forks (*Alzu et al., 2012*). Here we asked whether Senataxin is recruited to R-loops in the absence of CtIP or XPG and found that depletion of either factor increases Senataxin occupancy at the inducible (Sγ3) locus (*Figure 6G*) by 3 to 5-fold, similar to our observations of Sen1 in yeast.

## CtIP depletion leads to fewer DNA breaks after CPT treatment

Our results suggest that the CtIP and XPG nucleases help to either prevent R-loop formation in the genome or to resolve R-loops once they are formed. Since CtIP and XPG are both specific for 5' flaps, we considered a model in which CtIP and XPG process 5' flaps present in an R-loop structure (*Figure 7*). We hypothesize that this ssDNA cleavage event would result in extension and stabilization of the RNA-DNA hybrid, because the release of tension in one DNA strand would prevent spontaneous extrusion of the RNA. This conversion of the nascent lesion into an extended structure would also generate access for helicases to recognize and remove the RNA strand from the DNA. It is also possible that nuclease processing could convert the pre-lesion into double-strand breaks.

One prediction of this model is that CtIP and XPG-depleted cells would exhibit fewer single-strand DNA breaks compared to normal cells, even though they exhibit higher levels of R-loops. To test this idea, we created DNA damage with CPT and measured single-strand DNA breaks with an alkaline comet assay. We found that exposure to CPT generated a marked increase in DNA breaks, and that transcription promotes these breaks, as pretreatment with DRB reduced the levels of breaks by ~75% in wild-type cells (*Figure 8A*). CtIP depletion led to a significant reduction in the accumulation of DNA breaks after CPT treatment, consistent with the proposed model that CtIP can facilitate the conversion of stalled transcription associated DNA lesions into DNA breaks.

Next, we assessed the levels of single-strand DNA breaks in cells depleted for XPG, or for XPG and CtIP concurrently. In cells depleted for XPG there are fewer breaks after CPT treatment, and in cells depleted for both proteins, we observed no increase in breaks at all with CPT exposure (*Figure 8B*). This result does suggest that XPG and CtIP perform a function at transcription-associated lesions that results in an increase in single-strand breaks, likely their common function in promoting cleavage of 5' flaps. To test this, we also examined breaks by comet assay in cells expressing

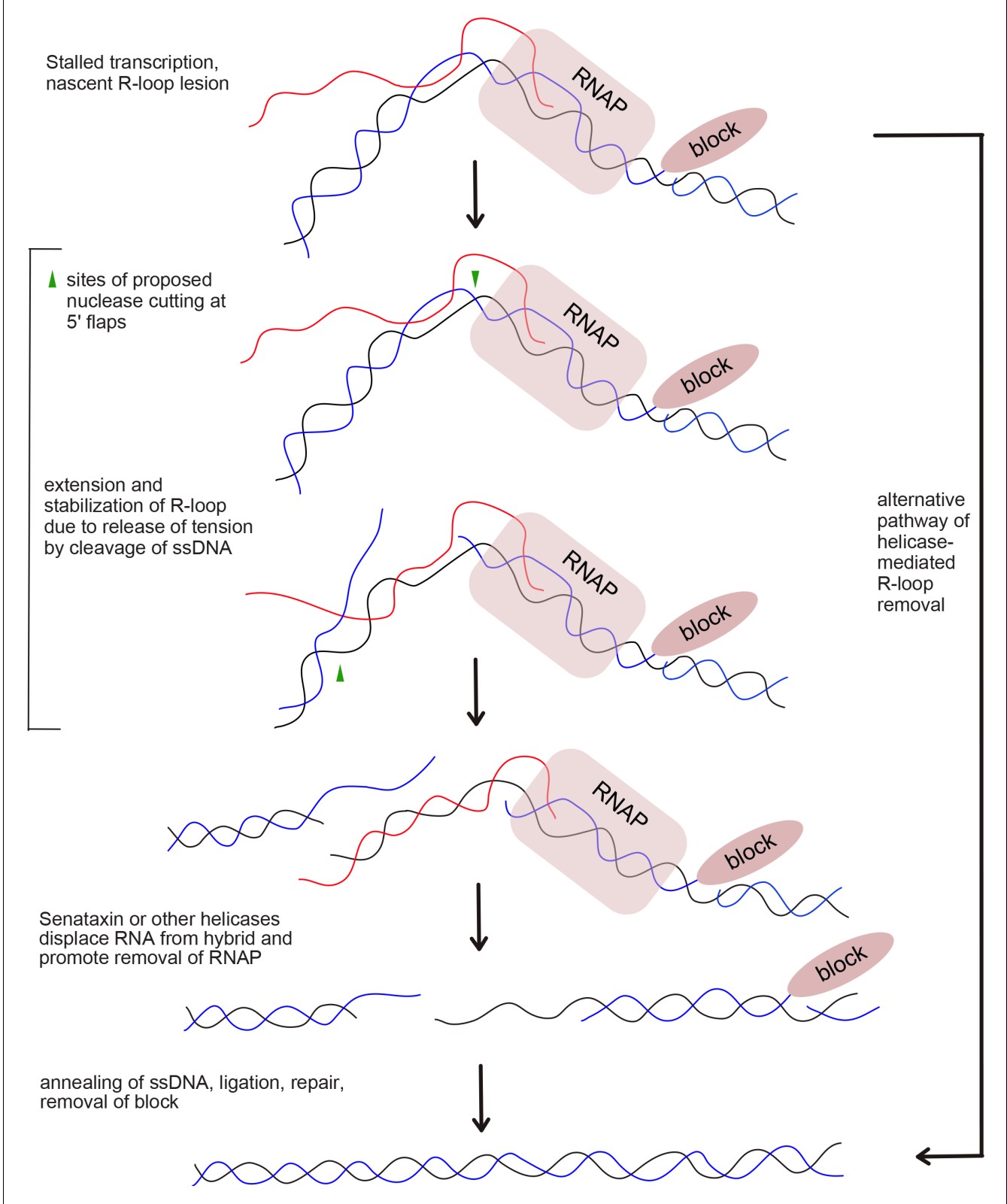

**Figure 7.** Proposed model of R-loop processing. Polymerase stalling at sites of nicks, topoisomerase adducts (TopI adduct shown here on non-template strand), or other lesions generates a nascent R-loop structure (RNA shown in red). Cleavage of this nascent structure at one of 2 exposed 5' flaps by nucleases, suggested in this work to be CtIP or XPG, can generate a stabilized, extended RNA-DNA hybrid because of the release of torsional constraint in the DNA template. Senataxin (or other helicases) preferentially access the RNA-DNA hybrid in this stabilized intermediate and remove the

*Figure 7 continued on next page*

*Figure 7 continued*

RNA, promoting reannealing of the displaced non-template strand and single-strand DNA repair. In this hypothetical model we do not know if one or both strands are cut, the timing or regulation of RNA polymerase removal, or the exact regulation of Senataxin activity at the lesion. A theoretical, alternative pathway of Sae2/CtIP-independent R-loop removal involving Senataxin is shown at right.

DOI: https://doi.org/10.7554/eLife.42733.016

the nuclease-deficient NA/HA CtIP mutant and found that in this case also, there was a large reduction in breaks formed with CPT exposure, very similar to the uncomplemented cells (*Figure 8C*).

To further validate this result at a specific locus in the genome, we returned to the RPL13A gene, where we observed high levels of R-loops in cells lacking CtIP using DRIP-qPCR (*Figure 6*). Here we used a modified LM-PCR assay (*Hatchi et al., 2015*) to measure single-strand DNA breaks in the genome. Primer extension from a sequence within the RPL13A locus converts any single-strand breaks present into single-ended double-strand breaks, then ligation-mediated PCR followed by qPCR quantitates the levels of these single-ended breaks (*Figure 8D*). Using this procedure, we did observe spontaneous single-strand breaks on the template strand of the RPL13A gene, and the level of breaks was reduced by 40% to 50% with depletion of CtIP, again consistent with the observation that single-strand breaks are reduced in cells lacking CtIP.

## Discussion

In this study we demonstrate a functional relationship between the Sae2/CtIP enzyme and transcription-related DNA damage in eukaryotic cells, showing that the damage sensitivity of cells lacking Sae2/CtIP can be rescued by either Sen1/Senataxin or RNaseH overexpression. We further provide evidence for accumulation of stalled RNA polymerase complexes and RNA-DNA hybrids in cells deficient in Sae2/CtIP, with these lesions located at sites of highly expressed genes. These results suggest that Sae2/CtIP is not only a double-strand break resection factor, but also functions at transcription-associated lesions in eukaryotic cells.

In *S. cerevisiae*, the temperature-sensitive *sen1-1* mutant exhibits synthetic lethality in combination with DNA repair mutants *mre11*, *rad50*, *sgs1*, and *rad52*, indicating a requirement for homologous recombination in the absence of *SEN1* function (*Mischo et al., 2011*). Consistent with these results, we observe a synthetic sensitivity of *sen1-1* with *sae2* for survival of CPT and MMS exposure, similar to the sensitivity of *sae2 rnh1 rnh201* strains. Conversely, overexpression of *SEN1, RNH1*, or *PCF11* partially rescues *sae2Δ* survival of these DNA damaging agents.

From *sen1* separation of function mutants we conclude that the helicase activity of Sen1 is important for the effect on survival of *sae2Δ* strains to DNA damage. Since we also observe stalling of RNA polymerase in *sae2Δ* strains, particularly with CPT treatment, one possible explanation is that Sae2 promotes the processing or resolution of transcription complexes stalled by Top1 conjugates and that *SEN1* overexpression rescues survival in this context because of its known activities in removing RNA-DNA hybrids. Our observations that *RNH1* overexpression has effects similar to *SEN1* indicate that the removal of the ribonucleotides from DNA is an important aspect of the suppression. Considering that we observe Sen1 at sites of transcription-related lesions in *sae2Δ* strains, it is conceivable that Sen1 acts in a parallel pathway to that of Sae2, and overexpression of this pathway promotes RNA removal and thus survival (*Figure 7*). Alternatively, it is possible that Sen1 plays a role downstream of Sae2 and that in the absence of Sae2 function, Sen1 catalytic activity is less efficient even though its recruitment to damage sites appears to be unimpeded.

In human cells, overexpression of Senataxin also complements CtIP-depleted cells for CPT survival and for accumulation of R-loops at sites of laser-induced DNA damage. The extent of this suppression is remarkable and suggests that R-loops are a limiting factor in the recovery of CtIP-depleted cells following DNA damage. In mammalian cells, loss of CtIP is lethal whereas yeast strains lacking *SAE2* have no growth deficit in the absence of DNA damage (*McKee and Kleckner, 1997*; *Prinz et al., 1997*; *Chen et al., 2005*; *Makharashvili and Paull, 2015*). This could be related to our observation that CtIP-depleted human cells exhibit higher levels of spontaneous R-loops than wild-type cells, even in the absence of exogenous damage, whereas *sae2* null yeast strains only show RNA polymerase II pausing and R-loop accumulation with damage treatment.

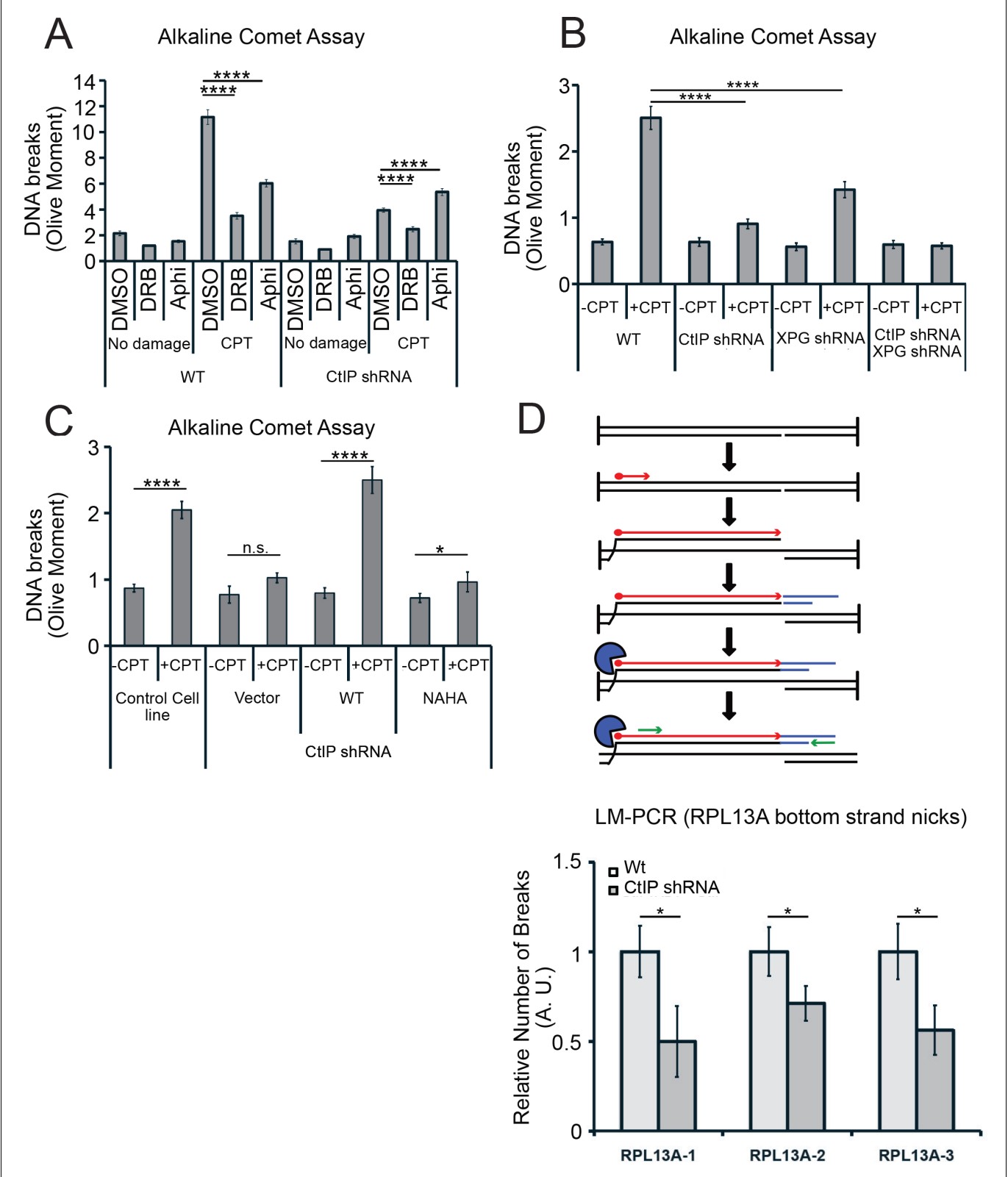

**Figure 8.** CtIP and its nuclease activity promote ssDNA break formation. (**A**) DNA breaks were quantified in wild-type and CtIP-depleted U2OS cells by alkaline comet assay. Cells were untreated (DMSO) or exposed to 5 µM CPT for 1 hr, with DRB (20 µM) or aphidicolin (2 µg/mL) pretreatment as indicated. Olive moments were calculated by analyzing at least 100 comets for each sample; error bars represent S.E.M. (**B**) Quantification of ssDNA breaks by alkaline comet assay was performed in wild-type, CtIP-depleted, XPG-depleted, or CtIP/XPG-depleted U2OS cells as in (**A**). (**C**) Quantification

*Figure 8 continued on next page*

*Figure 8 continued*

of ssDNA breaks by alkaline comet assay was performed in wild-type or CtIP-depleted U2OS cells complemented with wild-type eGFP-CtIP or nuclease-deficient NA/HA mutant CtIP as in (**A**). (**D**) A schematic representation of Ligation-mediated (LM)-PCR assay (top). Primer extension with a biotinylated primer (in red) from genomic DNA produces a double-stranded DNA end that is isolated with streptavidin and amplified by ligation-mediated PCR (asymmetric duplex and nested primers are presented in blue and green, respectively). LM-PCR assay measuring DNA breaks on the bottom strand of the RPL13A gene (bottom). DNA single-strand breaks were measured by LM-PCR at the endogenous RPL13A locus in wild-type or CtIP-depleted cells; n = 6, error bars represent S.E.M. * and **** denote p < 0.05 or 0.0001, respectively, using Student's two-tailed T test with comparisons as indicated.

DOI: https://doi.org/10.7554/eLife.42733.017

CtIP and Sae2 exhibit an intrinsic endonuclease activity that is specific for 5' flap structures and can be genetically separated from its ability to stimulate the nuclease of activity of Mre11 (*Makharashvili et al., 2014*; *Wang et al., 2014*; *Arora et al., 2017*). In human cells we observed that a CtIP mutant deficient in endonuclease activity failed to rescue CtIP-depleted cells for reduction of R-loops at laser damage sites or at genomic loci, thus we conclude that the action of CtIP at sites of stalled transcription involves its nuclease activity. Since we also show in this work that R-loop formation in CtIP-depleted cells resembles that of cells depleted of XPG, an endonuclease that also acts in nucleotide excision repair, it is possible that both enzymes target 5' flap structures, of which there are at least two present in every R-loop structure (see model in *Figure 7*). Evidence for a DNA cleavage event induced by CtIP or XPG in response to CPT or other transcription stalling lesions comes from our analysis of CPT-treated cells using alkaline comet assays, and also by locus-specific quantitation of single-strand breaks. These results clearly show that there are fewer single-strand breaks after DNA damage in the absence of CtIP or XPG, even though the levels of R-loops in these cells are much higher. We do not know if there is a specificity for either template or non-template ssDNA breaks for these enzymes, but we do observe a reduction in template strand breaks at the RPL13A locus with depletion of CtIP.

Although R-loops as measured by RNaseH-binding and S9.6 antibody immunoprecipitation are significantly higher in CtIP or XPG-depleted cells, depletion of both factors simultaneously shows a very different result: R-loop levels similar to wild-type cells. We observed this phenomenon in untreated cells, in cells treated with CPT, and in cells with laser-induced DNA damage. To explain this finding we propose that, in the absence of both CtIP and XPG, unprocessed nascent R-loops and/or stalled RNA polymerase complexes accumulate (*Figure 7*). We propose that, due to the topological constraints on R-loop formation, these nascent R-loop structures are small and are not efficiently recognized by either the S9.6 antibody or by RNase H. We know that there are toxic lesions present in cells depleted of both CtIP and XPG since the survival of cells depleted for both factors is very poor (*Figure 4—figure supplement 1*). Alternatively, it is possible that the transcriptional patterns change in cells depleted of both CtIP and XPG and that R-loops are much lower in this mutant for this reason. We cannot exclude this possibility with the data currently available, but it is certainly testable in future experiments.

The idea that a nascent R-loop is topologically constrained and relatively inaccessible in the absence of strand breaks is consistent with previous work showing that a nick in the non-template strand greatly improves the efficiency of stable R-loop formation, because the non-template strand then does not compete with the RNA for hybridization to the template strand (*Roy et al., 2010*; *Aguilera and Gómez-González, 2017*). Here we postulate that transient stabilization of an RNA-DNA duplex could initiate R-loop formation but that extensive spreading and stabilization of the hybrid would require either secondary structure formation in the non-template strand or cleavage of one of the DNA strands, as previously shown (*Roy et al., 2010*). We propose that cleavage of one of the DNA strands, although seemingly detrimental in its stabilization of the RNA-DNA hybrid, likely facilitates removal of the hybrid by transcription-associated helicases such as Senataxin and/or Aquarius. Biochemical characterization of the helicase domain of yeast Sen1 showed that the enzyme requires single-stranded DNA adjacent to the hybrid for efficient loading prior to unwinding of the annealed RNA (*Martin-Tumasz and Brow, 2015*). This type of structure is not available in an R-loop in a topologically closed system without cleavage of a DNA strand or active unwinding of the DNA duplex adjacent to the RNA-DNA hybrid. In addition, a recent study found that treatment of chromatin with low levels of the single-strand DNA-specific S1 nuclease greatly improves the yield of

R-loops immunoprecipitated by the S9.6 antibody (*Wahba et al., 2016*), consistent with this hypothesis.

There are also alternative hypotheses about the nature of the initiating lesion at genomic sites that require Sae2/CtIP. One possibility is that double-strand breaks form at sites of transcription-replication collisions, since we know that R-loops greatly increase the level of genomic instability at locations where they accumulate, and double-strand breaks have been implicated in these events (*Richard and Manley, 2017*; *Wickramasinghe and Venkitaraman, 2016*). Recent work demonstrates that R-loops form in human cells at DNA sites adjacent to induced double-strand breaks and that the Senataxin enzyme promotes their removal as well as Rad51 recruitment and homologous recombination (*Cohen et al., 2018*). In addition, a subset of R-loops in yeast were shown to induce Rad52-bound double-strand breaks, with the RNA-DNA hybrid located on one side of the break and resected DNA on the other (*Costantino and Koshland, 2018*). It is conceivable that CtIP and XPG promote the activity of Senataxin at such lesions, removing R-loops to promote resection. Lastly, double-strand breaks have been shown in some contexts to accompany high levels of transcriptional activity and appear to facilitate gene expression (*Madabhushi et al., 2015*); in this case TopII was implicated in formation of the breaks, but it is not clear if processing of the breaks occurs at these sites or is important for cell survival.

It is also possible that the initiating lesion is simply a stalled RNA polymerase, and that processing of this structure to create single-strand breaks first generates a transient R-loop as described above, but ultimately promotes the release of the polymerase and prevents the formation of stable, extensive R-loops. This idea that the polymerase is still present in the initiating lesion is supported by our data in yeast showing increased occupancy of RNAPII in *sae2Δ* strains that is reduced by *Sen1* overexpression. PCF11 also reduces the sensitivity of *sae2Δ* strains to DNA damaging agents, and PCF11 is known for its activities in promoting transcription termination, although it also promotes Sen1 recruitment (*Grzechnik et al., 2015*).

In conclusion, we have presented evidence supporting a role for Sae2/CtIP in processing of transcription-related DNA lesions and show that release of R-loops from the genome is a limiting factor for the survival of Sae2/CtIP-deficient cells following DNA damage. This is perhaps related to the fact that CtIP was first identified through its association with the transcription regulator CtBP (*Schaeper et al., 1998*), and also interacts directly with the tumor suppressor BRCA1, which associates with transcription-related complexes (*Makharashvili and Paull, 2015*; *Yu et al., 1998*; *Chinnadurai, 2006*). BRCA1 itself has been shown to localize to a subset of transcription termination regions and to regulate levels of RNA-DNA hybrids in human cells (*Hatchi et al., 2015*; *Hill et al., 2014*). In addition, XPG was recently found to be present in a complex with BRCA1 that is important for cellular responses to DNA damage that is separate from its roles in excision repair (*Trego et al., 2016*). Further studies will need to address whether the roles of CtIP and BRCA1 at transcription-associated lesions are functionally interdependent and whether Senataxin can also rescue BRCA1-deficient cells in the same manner as CtIP-depleted cells. Lastly, CtIP has been shown to have important repair functions in $G_1$ phase cells (*Barton et al., 2014*; *Biehs et al., 2017*; *Helmink et al., 2011*; *Quennet et al., 2011*; *Yun and Hiom, 2009*); it will be important to determine if the R-loop processing functions indicated here play a role in the repair of transcription-associated damage even outside of S phase and whether this accounts for any of the essential functions of CtIP in mammals.

## Materials and methods

**Key resources table**

| Reagent type (species) or resource | Designation | Source or reference | Identifiers | Additional information |
|---|---|---|---|---|
| Strain, strain background (yeast) | yeast strains see *supplementary file 1* | | | |
| Cell line (human) | U2OS | ATCC | | |
| Cell line (human) | HEK-293T | ATCC | | |

*Continued on next page*

*Continued*

| Reagent type (species) or resource | Designation | Source or reference | Identifiers | Additional information |
|---|---|---|---|---|
| Cell line (human) | U2OS T-RexTM Flp-in | Jeff Parvin | U2OS Flp-in | Invitrogen tet-on system |
| Transfected construct (human) | pTP3146 | this study | GFP-CtIP | see Materials and methods |
| Transfected construct (human) | pTP3663 | this study | Flag-Bio-CtIP | see Materials and methods |
| Transfected construct (human) | pTP3665 | this study | Flag-Bio-CtIP (N289A/H290A) | see Materials and methods |
| Transfected construct (human) | pTP3531 | this study | C-term of Senataxin in pcDNA5 /FRT/TO | see Materials and methods |
| Transfected construct (human) | pTP4195 | this study | PCF11 in pcDNA5/ FRT/TO | see Materials and methods |
| Transfected construct (E. Coli) | pTP3492 | this study | wild-type RNaseH-mCherry in pcDNA5/FRT/TO | see Materials and methods |
| Transfected construct | pTP3494 | this study | mCherry only in pcDNA5/FRT/TO | see Materials and methods |
| Transfected construct (E. Coli) | pTP3493 | this study | RNaseHI -D10R- E48R in pcDNA5/ FRT/TO | see Materials and methods |
| Transfected construct (E. Coli) | pTP3660 | this study | RNaseHI -D10R- E48R in lentivirus vector | see Materials and methods |
| Transfected construct | pRSITEP–U6Tet-(sh)-EF1 -TetRep-2A-shRNA-CtIP | Cellecta | inducible shRNA for CtIP | see Materials and methods |
| Transfected construct | pTP3703 | this study | inducible shRNA for XPG | see Materials and methods |
| Transfected construct | pTP3677 | this study | inducible shRNA for SETX | see Materials and methods |
| Transfected construct | pTP4195 | this study | Sγ3 sequence in pcDNA5/FRT/TO | see Materials and methods |
| Other | primers see **supplementary file 4** | | | |

## *S. cerevisiae* strains and expression constructs

For yeast strains see **supplementary file 1**. Genomic deletions were made with standard lithium chloride transformations and integrations (*Adams et al., 1997*) and were verified by PCR. Full-length *SEN1*, *SSU72*, and *RTT103* were cloned into the 2μ plasmid pRS425 (*Christianson et al., 1992*) by fusion PCR to create pTP3500, pTP3249, and pTP3498, respectively. A pRS425 derivative containing *PCF11* was a gift from Steve Hanes. Mutations in *SEN1* were made in pTP3500 by Quikchange muta-genesis (Stratagene). The *sen1-1* strain and corresponding wild-type strain were gifts from Nicholas Proudfoot. The HTB-tagged Rpb2 and Sen1 strains were gifts from Jeff Corden. The rnh1 rnh201 mutant strain was a gift from David Tollervey. A high-copy vector containing the wild-type *SAE2* gene with a 2xFLAG tag in pRS425 (61)'*FLAG-SAE2/2μ*' was a gift from John Petrini, and was used for the *SAE2* ChIP experiment.

## Yeast camptothecin survival assay

Cells were grown to exponential phase (OD 0.5–0.6) and synchronized using alpha-factor for 3 hr at 30°C. Half of the cells were kept in alpha factor ($G_1$), while the other half were washed and resuspended in fresh media (S). Previous studies have shown that all the cells enter S phase 30 min after the removal of alpha factor (*Fu et al., 2014*). Therefore 30 min after the release into fresh media, both $G_1$ and S samples were treated with either 100 µM CPT or DMSO for two hours for an acute drug dose. At the end of the treatment, cells were washed and appropriate dilutions were plated on YPDA +glucose plates to obtain single colonies. Single colonies were counted after two days and survival efficiency was calculated as the number of colonies on the '+CPT' plates divided by the number of colonies on the 'no treatment' plates. For thiolutin treatment, 15 min after the release into fresh media both $G_1$ and S phase cells were treated with either DMSO or thiolutin (2.5 µg/mL final concentration). After another 15 min CPT or DMSO was added to the media at indicated concentration.

## Chromatin immunoprecipitation in yeast

Sae2: Yeast cells containing a FLAG-Sae2 expression cassette on a high copy number 2µ plasmid were grown to exponential phase in 2 L of appropriate minimal media to $OD_{600}$ = 0.5–0.6. The cells were centrifuged and resuspended in 50 ml fresh media (concentrated 40-fold). Yeast mating pheromone alpha-factor was added to final concentration of 10 µM and cells were synchronized for 4 hr at 30°C. Half of the cells were kept in alpha factor ($G_1$), while the other half were washed and resuspended in fresh media (S). After the wash, both $G_1$ and S phase cells were further divided into two and treated with 100 µM camptothecin or DMSO for 50 min. At the end of the drug treatment cells were crosslinked by addition of formaldehyde (1% final concentration) at RT for 25 min, followed by glycine (125 mM final concentration) at RT for 5–10 min. The cells were harvested, washed with water and flash frozen until further processing. Thus 2L of starting culture was divided into four samples per strain with approximately 500 OD cells per sample.

Each cell pellet was resuspended in 400 µl lysis buffer (50 mM HEPES, 150 mM NaCl, 1 mM EDTA, 1 mM DTT, 10% glycerol, 0.1% NP40, and protease inhibitors (Pierce # A32955; 1 tablet per 10 ml) and lysed using a bead beater (3 cycles of 45 s) in the presence of 400 µl 0.5 mm zirconia beads. The beads were washed with 2 × 500 µl of lysis buffer to collect a total of 1.5 ml cell extract. Sonication was performed with a Branson Digital Sonifier (5 cycles of 1 min each, with 10 s on, 10 s off at 28% amplitude). The cells were kept on ice for 5 min in between each cycle. The extract was cleared with high-speed centrifugation at 4°C for 30 min. A small sample was treated with proteinase K and RNaseA and separated by agarose gel to check for sonication efficiency (DNA size should be between 100–250 bp). The lysate was diluted to 10 ml with the lysis buffer. A 500 µl aliquot of the lysate was saved as the 'Input' sample. 50 µg Sigma FLAG-M2 antibody and 300 µl Pierce Protein A/ G magnetic beads (prewashed with lysis buffer) were added to the cleared lysate. The cells were rotated at 4°C overnight. The following day, the beads were separated and washed 3 times each with wash-buffer 1 (25 mM Tris pH 8, 1 mM EDTA, 150 mM NaCl, 1% Triton, 100 µM camptothecin or DMSO equivalent), wash buffer 2 (25 mM Tris pH = 8, 1 mM EDTA, 500 mM NaCl, 1% Triton), wash buffer 3 (10 mM Tris pH 8, 1 mM EDTA, 1% Triton, 500 mM LiCl, 0.5% NP40) and wash buffer 4 (25 mM Tris pH 8, 1 mM EDTA, 1% Triton, 150 mM NaCl). 500 µl elution buffer (wash buffer 4 + 3X FLAG peptide) was added to the beads and kept shaking at 4°C for 2 hr. The beads were eluted with 100 µl elution buffer (wash buffer 4 + 2% SDS) and incubated at 65°C for 30 min. A second elution was done with 100 µl elution buffer similarly and combined for 150 µl total elution. The final elution and the Input sample (supplemented to 2% final SDS) were reverse cross-linked at 65°C overnight. Next day, all samples were diluted to SDS <0.5% and treated with 20 µg RNase A at 37°C 2 hr followed by 40 µg of proteinase K at 37°C 2 hr. The DNA was purified by phenol chloroform extractions followed by ethanol precipitation and stored at −20°C until further processing.

The DNA samples were gel purified to obtain a size range of 50–200 bp. The sequencing libraries were prepared for each sample using NEBNext DNA Library prep kit for Illumina (E6040S) and were sequenced and analyzed at the UT Austin GSAF. The bam files and bed files were visualized using Integrated Genome Viewer (iGV) and compared with saccer3 reference genome. Primary data is available at the NCBI GEO website under GSE122782.

*RPB2*: Yeast cells containing a His6-Tev-Biotin (HTB)-tagged *RPB2* allele in the genome were used for this experiment. Cell pellets were prepared, lysed and sonicated similar to Sae2 ChIP, except the following lysis buffer was used - 25 mM Tris pH 7.5, 150 mM NaCl, 1 mM EDTA, 1 mM DTT, 10% glycerol, 0.1% NP40, 0.1% SDS, 1% Triton, 200 mM PMSF supplemented with protease inhibitors (Pierce # A32955; 1 tablet per 10 ml). After sonication the lysate was divided into three parts - 10% for Input, 45% for Rpb2 IP and 45% for mock IP. The IP was done with streptavidin linked M280 magnetic beads (Invitrogen) overnight and sheep anti-mouse M280 magnetic beads (Invitrogen) were used for the mock. The next day the beads were separated and washed 3 times with the following buffers - 1 (25mM Tris pH 7.5, 150 mM NaCl, 1 mM EDTA, 1% Triton, 0.1% SDS), 2 (25 mM Tris pH 7.5, 500 mM NaCl, 1 mM EDTA, 1% Triton, 0.1% SDS), 3 (10 mM Tris pH 8, 500 mM LiCl, 1 mM EDTA, 1% Triton, 0.5% NP40), and 4 (10mM Tris pH 8, 1 mM EDTA). For each wash, the beads were rotated for 5 min at RT. The fold enrichment over Input at different genomic loci was calculated using SYBR-Green based quantitative PCR. Samples were tested at various dilution factors to optimize a dilution factor for a CT value in the linear range. Then percentages of DNA enrichment of IP over the Input were calculated by using the following equation:

$$(100\% * 2^{(CT_{(input)} - CT_{(IP)} - \log_{(dilution\_factor, 2)})} - 100\% * 2^{(CT_{(input)} - CT_{(Mock\ IP)} - \log_{(dilution\_factor, 2)})}).$$

Then relative fold of enrichment was calculated by normalizing the percentage of DNA enrichment of IP of each group to WT cells without CPT treatment.

*SEN1*: Yeast cells containing a His6-Tev-Biotin (HTB)-tagged *SEN1* allele in the genome were prepared, lysed and sonicated similar to *RPB2* ChIP except that all samples were prepared with CPT exposure (100 µM camptothecin or DMSO for 50 min). The IP was performed with streptavidin linked M280 magnetic beads (Invitrogen). The fold enrichment over Input at different genomic loci was calculated using SYBR-Green based quantitative PCR. Samples were tested at various dilution factors to optimize a dilution factor for a CT value in the linear range. Then percentages of DNA enrichment of IP over the Input were calculated by using the following equation: $(100\% * 2^{(CT_{(input)} - CT_{(IP)} - \log_{(dilution\_factor, 2)})})$. The relative fold of enrichment was calculated by normalizing the percentage of DNA enrichment of IP of each group to WT cells.

## DNA-RNA ImmunoPrecipitation (DRIP) in yeast

Yeast cells were grown as described for the Sae2 ChIP experiment, except there were no G2 cell samples and no formaldehyde crosslinking was done. DRIP was performed as previously described (*Boguslawski et al., 1986*); a detailed protocol is available on request. Briefly, 150–200 µg genomic DNA isolated using the Qiagen Genomic DNA kit (500G) was treated with S1 nuclease and precipitated in 130 µl TE. The DNA was sonicated using a Covaris sonicator, precipitated, and resuspended in 50 µl of nuclease-free water. Half of the sample was treated with RNaseH in vitro (Thermo Rnase H, 5 units per 10 µg DNA), overnight at 65°C. Then 350 µl of FA buffer (1% Triton X-100, 0.1% sodium deoxycholate, 0.1% SDS, 50 mM HEPES, 150 mM NaCl, 1 mM EDTA) was added to each DNA sample, and incubated for 90 min with 5 µg of S9.6 antibody prebound to magnetic protein A beads. Beads were then washed and the DNA eluted as described above. %RNA–DNA hybrid amounts were quantified using Sybr-Green based quantitative PCRs on DNA samples from DRIP and Input DNA. Q-PCR reactions were performed on ViiA7 Real-Time PCR System (ABI) under standard thermal cycling conditions for 40 cycles. Results were analyzed with ViiA7 software (ABI). For each sample, fold enrichment over Input was calculated by the following equation: Fold enrichment over input = $2^{(Ct^{Input} - Log(dilution\ factor, 2) - Ct^{DRIP})}$

Primer sequences used in this study were adapted from (*Alzu et al., 2012*; *Grzechnik et al., 2015*; *El Hage et al., 2014*) and are listed in *Supplementary file 4*. A mock IP with empty beads was done for each sample to control for non-specific binding and enrichment was calculated in the same way as the DRIP sample.

## Statistical analysis of Sae2-ChIP-transcription overlap

For each sample, overlap between peaks identified from ChIP-seq data and the top 10% of highly transcribed genes in yeast (*supplementary file 2*) (*Nagalakshmi et al., 2008*; *Pelechano et al., 2010*; *Miura et al., 2008*) was assessed using Bedtools (*Quinlan and Hall, 2010*) and the fraction of peaks for which an overlap was detected was recorded. Also for each sample a background null distribution of overlap rates was estimated by repeatedly sampling a random set of coding sequences

equal in number to those in the top 10% list from the Saccharomyces cerevisiae genome (equivalent to selection of top ORFs after random permutation) and then locating overlaps using bedtools. By comparing the overlap fraction to the estimated null distribution, we constructed 99% confidence intervals for the permutation test p-values (*Ernst, 2004*) of the null hypothesis that the overlap between ChIP-seq peaks and the top 10% of highly transcribed genes was equal to the rate at which the peaks overlapped randomly chosen coding sequences.

*Mammalian cell culture:* Human U2OS and HEK-293T cells were grown and maintained in tetracycline-free DMEM containing 10% FBS media in a humidified 37°C incubator in the presence of 5% $CO_2$. All cell lines were treated and maintained in plasmocin (InVivoGen) to ensure no mycoplasma contamination.

## Mammalian expression constructs

Invitrogen Gateway pENTR223 donor vector for hSenataxin(Δ1–1850) was obtained from DNASU (#HsCD00505781), and cloned into pcDNA5/FRT/TO (ThermoFisher) vector to make pTP3531. shRNA CtIP was custom-made by Cellecta (pRSITEP–U6Tet-(sh)-EF1-TetRep-2A-shRNA-CtIP) and includes the shRNA expression cassette 5'-GAGCAGACCTTTCTTAGTATAGTTAATATTCATAGCTATACTGAGAAAGG

TCTGCTCTTTT-3 '. Dox-inducible elements were removed from pRSITEP–U6Tet-(sh)-EF1-TetRep-2A-shRNA-CtIP shRNA CtIP(-Dox) (pTP3914) using Q5 Site-Directed Mutagenesis (NEB, #E0554S). The N-terminal fusion of eGFP with CtIP (eGFP-CtIP) ORF was amplified from pC1-eGFP-CtIP (a generous gift from Steven Jackson), and cloned into pcDNA5/FRT/TO to make pTP3146. The A206K mutation in eGFP was made to prevent eGFP dimerization, and shRNA resistance mutations introduced to generate pTP3148. pcDNA5-flag-bio-CtIP(wt) (containing N-terminal Flag and biotinylation signal sequences) (pTP3663) was made by Q5 Site-Directed Mutagenesis to remove eGFP, and insert Flag and biotinylation signal sequences. pcDNA5-flag-bio-CtIP(N289A/H290A) (pTP3665) was made using QuikChange II Site-Directed Mutagenesis (Agilent Technologies). shRNA SETX1 (pRSI-TEP–U6Tet-(sh)-EF1-TetRep-2A-HYGRO) (pTP3677) and shRNA XPG (pRSITEP–U6Tet-(sh)-EF1-TetRep-2A-HYGRO) (pTP3703) were made from the pRSITEP–U6Tet-(sh)-EF1-TetRep-2A-shRNA-CtIP construct (Cellecta) using Q5 Site-Directed Mutagenesis, with the shRNA cassettes 5'-GCCAGATCGTATACAATTATAGTTAATATTCATAGCTATAATTGTATACGATCTGGCTTTT-3 ' and 5'-GAACG-CACCTGCTGCTGTAGAGTTAATATTCATAGCTCTACAGCAGCAGGTGCGTTCTTTT-3 ', respectively. pcDNA5 with with wild-type RnaseHI (rnhA from *E. coli*) containing an NLS and fused to mCherry (pTP3492) was generated from pcDNA5/FRT/TO (Invitrogen) and pICE-RNaseHI-mCherry (Addgene 60365, gift from Patrick Calsou). pcDNA5 with NLS-mCherry (pTP3494) was generated from pcDNA5/FRT/TO (invitrogen) and pICE-mCherry (Addgene # 60364). pcDNA5-RNaseHI-D10R-E48R-NLS-mCherry (pTP3493) was cloned using pcDNA5/FRO/TO and pICE-RNaseHI[D10R-E48R]-NLS-mCherry (a gift from Patrick Calsou, Addgene # 60367 (*Britton et al., 2014*)). pLenti-PGK-RNase-HI[D10R-E48R]-NLS-mCherry (pTP3660) was cloned using pLenti PGK GFP Blast (w510-5) which was a gift from Eric Campeau and Paul Kaufman (Addgene plasmid #19069 [*Campeau et al., 2009*]) and the RNaseHI[D10R-E48R]-NLS-mCherry fragment from pICE-RNaseHI[D10R-E48R]-NLS-mCherry. pBluescript-Sγ3 × 12 (pTW-121) was a generous gift from Michael Lieber. pcDNA5/FRT/TO-Sγ3 × 12 (pTP4195) was cloned using pcDNA5/FRT/TO and the Sγ3 region from pTW-121. pcDNA5/FRT/TO-hPCF11 (pTP4195) was cloned using pcDNA5/FRT/TO and the hPCF11 ORF containing plasmid, obtained from Kazusa (ORK06290). All constructs and mutations were confirmed by DNA sequencing. Details of plasmid construction available upon request.

## Clonogenic survival assays

U2OS cells were harvested by trypsinization (0.25% trypsin; Life Technologies) and counted using Scepter (Millipore Sigma). 1,500 cells were seeded per 10 cm cell culture dish, and were allowed to adhere to the bottom of the plate for 36 hr. For experiments with tet-inducible CtIP shRNA, all cells were exposed to doxycycline (1 μg/mL) during seeding and gene depletion and/or over-expression and throughout the experimental course. After 36 hours cells were treated CPT for 1 hr, the CPT-containing media was replaced with fresh media, and cell recovery was allowed for 10 days. Colonies were stained with crystal violet (0.05% in 20% ethanol), destained with water, and counted using a scanner and Image J.

## RNaseH$^{D10R-E48R}$-mCherry Laser Micro-irradiation

U2OS cells were seeded in glass bottom petri dishes (35 × 10 mm, 22 mm glass, WillCo-dish, HBST-3522), and grown in DMEM/10% FBS media in the presence of 1 µg/mL doxycycline. After 36 hr, media was replaced with media containing 10 µM BrdU. After an additional 36 hr, laser micro-irradiation was performed with an inverted confocal microscope (FV1000; Olympus) equipped with a CO$_2$ module and a 37°C heating chamber. A preselected spot within the nucleus was microirradiated with 20 iterations of a 405 nm laser with 100% power to generate localized DNA damage. Then, time-lapse images were acquired using a red laser at 1 min time intervals for 10 min. The fluorescence intensity of mCherry signal at the laser microirradiated sites was measured using the microscope's software. Data collected from >10 cells were normalized to their initial intensity and plotted against time.

## RNaseH$^{D10R-E48R}$-mCherry FACS

U2OS cells were grown in DMEM/10% FBS media in the presence of 1 µg/mL doxycyclin in 10 cm dishes. After 3 days, the cells were harvested by trypsinization, rinsed in 5 mL cold PBS with Ca$^{2+}$ (0.9 mM) and Mg$^{2+}$ (0.5 mM), and centrifuged at 1000 g for 3 min. Unbound RNaseH$^{D10R-E48R}$-mCherry was extracted with 1 mL Triton X-100 buffer (0.5% Triton X-100, 20 mM Hepes-KOH (pH 7.9), 50 mM NaCl, 3 mM MgCl$_2$, 300 mM Sucrose) at 4°C for exactly 2 min and at 1300 g for 3 min. Cells were then rinsed twice in PBS at RT and fixed in 3.7% paraformaldehyde in PBS for 10 min at RT, rinsed twice in PBS at RT again, and stained with FxCycleTM FarRed stain (200 nM in PBS with 100 µg/mL RNaseA). Cells were kept in the staining solution at 4°C protected from light. The samples were analyzed in a flow cytometer without washing, using 633/5 nm excitation and emission collected in a 660/20 band pass or equivalent.

## S9.6 immunostaining

U2OS cells were seeded into 8-well Nunc Lab-Tek II Chamber Slides (Nalge Nunc International, #154534) 48 hr before experiments. Prior to immunostaining, cells were washed with PBS, and pre-extracted with incubation in CSK buffer (10 mM PIPES, pH 7.0, 100 mM NaCl, 300 mM sucrose, and 3 mM MgCl$_2$, 0.7% Triton X-100) twice for 3 min at room temperature. After preextraction, cells were washed with PBS and fixed with 2% paraformaldehyde for 10 min. Cells were then permeabilized for 5 min with PBS/0.2% Triton X-100, washed with PBS, and blocked with PBS/0.1% Tween 20 (PBS-T) containing 5% BSA. For immunostaining cells were incubated with primary antibodies (S9.6 and nucleolin) in PBS/5% BSA overnight, then washed with PBS-T and incubated with appropriate secondary antibodies coupled to Alexa Fluor 488 or 594 fluorophores (Life Technologies) in PBS-T/5% BSA. After washes in PBS-T and PBS, coverslips were incubated 30 min with 2 µg/ml DAPI in PBS. After washes in PBS, coverslips were rinsed with water and mounted on glass slides using Pro-Long Gold (Life Technologies).

## DNA-RNA ImmunoPrecipitation (DRIP) in mammalian cells

U2OS cells (one 150 mm dish per biological replicate) were harvested by trypsinization (0.25% trypsin; Life Technologies) and pelleted at 1,000 g for 5 min in 15 mL conical tubes. Cell pellets were washed with PBS and divided for RNA, DRIP or LM-PCR harvests. Cell pellets for DRIP were resuspended in 5 mL of PBS supplemented with 0.5% SDS, and digested with 2 mg of Proteinase K (Gold-Bio) at 37°C overnight. Cell lysates were then extracted once with 1 vol of equilibrated phenol pH 8 and twice with 1 vol of chloroform. DNA was precipitated with 1 vol of isopropanol, and spun down at 6,500 g for 15 min. The DNA pellet was transferred to a 1.7 mL eppendorf tube, washed with 1 mL of 70% ethanol, and rehydrated in 0.1x TE (1 mM Tris-HCl pH 8.0, 0.1 mM EDTA). Nucleic acids were digested using a restriction enzyme cocktail (20 units each of EcoRI, HindIII, BsrGI, XbaI) (New England Biolabs) overnight at 37°C in 1x NEBuffer 2.1. DNA concentration was measured using Qubit dsDNA HS kit (Thermo). For experiments with RNaseH treatment in vitro, RNaseH (Thermo) was added at a concentration of 5 units per 10 µg DNA and incubated overnight at 37°C. 10 µg of digested nucleic acids were then diluted in 1 mL final DRIP buffer (10 mM sodium phosphate, 140 mM sodium chloride, 0.05% Triton X-100) and 100 µg of S9.6 antibody (purified from ATCC HB-8730) and incubated at 4°C overnight. This and all wash steps were performed on a rotisserie mixer. 30 µL of Pierce Protein A/G Magnetic Beads (Fisher Scientific, 88803) was added and incubated for

additional 2 hr, followed by washing three times with 1 mL of 1x IP buffer for 10 min at room temperature with constant rotation. After the final wash, the agarose slurry was resuspended in 100 µL of TE + 0.5% SDS 1 mg of Proteinase K for >1 hr at 37°C. 10 µL of 7.5 M Ammonium Acetate, 1 µg of glycogen, and 400 µL of 100% ice-cold Ethanol were added to the digested DRIP samples, and kept at −20°C for at least 2 hr (to overnight) to precipitate the immuno-precipitated material. The pellet was collected by centrifugation in a microcentrifuge at maximum speed for 30 min at 4°C, washed with cold 70% ethanol, air-dried, and resuspended in 100 µL of 0.1x TE. We used 10 µL reactions with PowerUp sybr green master mix (Applied Biosystems) for qPCR amplification of genomic loci (see *Supplementary file 4*). Reactions were incubated with the following program on a Viia 7 System cycler (Life Technologies): 50°C 2 min, 95°C 10 min, 40 cycles of 95°C 15 s, 64°C 1 min, followed by a melt curve: 95°C 15 s, 60°C 1 min, 0.05 °C/sec to 95°C 15 s. For each DRIP sample, linear range of amplification was identified by testing a wide range of dilutions. Fold enrichment for a given locus was calculated using the $2^{-\Delta\Delta CT}$ method (*Schmittgen and Livak, 2008*), and then normalizing the samples to the measurements of the wild-type results.

## RT-qPCR

U2OS cells (one 150 mm dish per biological replicate) were harvested by trypsinization (0.25% trypsin; Life Technologies) and pelleted at 1,000 g for 5 min in 15 mL conical tubes. Cell pellets were washed with PBS (Life Technologies) and divided for RNA, DRIP or LM-PCR harvests. RNA was either purified and retro-transcribed using RNA purification (Qiagen) and SuperScript IV Reverse Transcriptase (Thermo #8090050) kits, respectively, or using Fast Cells-to-CT kit (Thermo #4399003). qPCR was done using the same settings as those for DRIP-qPCR and LM-PCR methods, and GAPDH used as a reference gene.

## mRNA-seq

U2OS cells (one 150 mm dish per biological replicate) were harvested by trypsinization and pelleted at 1,000 RCF for 5 min in 15 mL Falcon tubes. mRNA was purified using Qiagen RNA purification kit, and mRNA was isolated using AMPure XP kit (Beckman Coulter, #A63880). mRNA-seq libraries were prepared with the NEBNext Multiplex Small RNA Library Prep Set for Illumina (#E7300S) according to manufacturer instructions. The library sequencing and analysis were done at New York Genome Center.

## Statistical analysis of mRNA-seq DRIP data overlap

For each statistical comparison, overlap between the top 100 genes as ranked by DESeq differential expression p-value and DRIPc-seq peaks from GEO dataset GSE70189 (Sanz 2016) was assessed using bedtools (*Quinlan and Hall, 2010*). For each comparison, we calculated the percentage of the top 100 genes for which such an overlap was found. These percentages were compared to a null distribution for overlap rates estimated using a permutation testing approach (*Ernst, 2004*) in which we repeatedly selected 100 genes at random from the list of all genes tested by DESeq (equivalent to selection of top 100 genes after randomly permuting DESeq p-values) and applied bedtools to calculate the DRIPc-seq overlap rate in the same manner. Using this approach we estimated 99% confidence intervals for the permutation test p-values of the null hypothesis that the genes showing the most evidence of differential expression overlapped the peak regions at the same rate as randomly selected genes.

## Senataxin ChIP-qPCR

U2OS cells (one 150 mm dish per biological replicate) were crosslinked by addition of formaldehyde (1% final concentration) at RT for 15 min, followed by glycine (125 mM final concentration) at RT for 5 min. Cells were washed twice with cold PBS and harvested by scraping. Then cells were pelleted at 3,000 rpm for 30 min in 15 mL Falcon tubes. Each cell pellet was resuspended in 2 mL of RIPA buffer (50 mM Tris pH8, 150 mM NaCl, 2 mM EDTA, 0.1% SDS, 0.5% Sodium Deoxycholate, 1% NP40, and protease inhibitors (Pierce # A32955; 1 tablet per 10 mL)) and sonicated with a Bioruptor sonicator for 15–30 min at high power 10 s on and 10 s off. The extract was cleared with 13,000 rpm centrifugation at 4°C for 30 s and the supernatant was transferred to new tube. A small sample was treated with 1% SDS, 100 mM NaHCO3 and RNaseA and purified by Qiagen PCR purification kit to

check DNA concentration with NanoDrop 2000 (Thermofisher). The chromatin was diluted to 50 µg/mL with RIPA buffer. A 50 µL aliquot of the lysate was saved as the 'Input' sample, and 1 mL of the lysate was used per immunoprecipitation sample. 2 µg of anti-SETX antibody (Novus Biologicals, NB100-57542) was added to all immunoprecipitation samples except the beads-only control and immunoprecipitated overnight with rotation at 4°C. 20 µL of Pierce Protein A/G Magnetic Beads (Fisher Scientific) was added and incubated for additional 2 hr, followed by washing 3 times each with wash buffer 1 (20 mM Tris pH8, 2 mM EDTA, 150 mM NaCl, 1% Triton, 0.1% SDS), wash buffer 2 (20 mM Tris pH8, 2 mM EDTA, 500 mM NaCl, 1% Triton, 0.1% SDS), wash buffer 3 (10 mM Tris pH8, 1 mM EDTA, 1% Sodium Deoxycholate, 250 mM LiCl, 1% NP40) and TE buffer (10 mM Tris pH8, 0.1 mM EDTA). 125 µl elution buffer (1% SDS, 100 mM NaHCO3) was added to the beads and kept shaking at 30°C for 30 min. The beads were then pelleted and the supernatant was transferred into fresh tube. The tube containing the supernatant was kept shaking overnight at 65°C. The DNA samples were purified (Qiagen PCR purification kit) and eluted with 50 µl of water (heated at 50°C and incubated for 30 min on column before spinning).

We used 10 µl reactions with PowerUp sybr green master mix (Applied Biosystems) for qPCR amplification of genomic loci (see *Supplementary file 4*). Reactions were incubated with the following program on a Viia 7 System (Life Technologies): 50°C 2 min, 95°C 10 min, 40 cycles of 95°C 15 s, 64°C 1 min, followed by a melt curve: 95°C 15 s, 60°C 1 min, 0.05 °C/second to 95°C 15 s. For each ChIP sample, linear range of amplification was identified by testing a wide range of dilutions. Fold enrichment for a given locus was calculated using the 2-ΔΔCT method (*Schmittgen and Livak, 2008*), and then normalizing the samples to the measurements of the control.

## Ligation-mediated PCR (LM-PCR)

U2OS cells (one 150 mm dish per biological replicate) were harvested by trypsinization and pelleted at 1,000 g for 5 min in 15 mL conical tubes. Cell pellets were washed with PBS and divided for RNA, DRIP or LM-PCR harvests. Genomic DNA (gDNA) was purified using genomic DNA preparation kit (Zymo Research Quick-gDNA MiniPrep - Capped column, Genesee Scientific, 11-317AC) and gDNA concentration was determined using Nanodrop. 50 µL primer extension mix contained: 5 µL of 10X polymerase buffer (NEB, supplied with Deepvent(-Exo) enzyme), 4 µL MgSO$_4$ (100 mM), 1 µL of Deepvent(-Exo) (NEB), 1 µL NTP mix (0.5 mM each final), 0.5 µL biotinylated primers (stock concentration 100 µM), and 1 µg of gDNA. Primer extension was done in a thermocycler in one round of primer extension: 15 min at 95°C, 30 s at 60°C, 5 min at 72°C. Control solution was made by dilution of 1 µg genomic DNA in 0.1x TE. Primer extension products were ligated to the phosphorylated asymmetric adaptor duplex overnight (oligonucleotides were phosphorylated with T4 polynucleotide kinase at 37°C for 3–4 hr, purified with a nucleotide removal kit (Qiagen), and annealed with boiling and slow cooling in the presence of 0.1 M NaCl). Ligation reactions were mixed with 30 µL of KilobaseBinder (Invitrogen) magnetic beads prepared according to the manufacturer's protocol, total volume was adjusted to 100 µL, and incubated with genomic DNA samples overnight. Washes were performed on a magnetic stand: 3 × 10 min washing with wash buffer (50 mM Tris, pH 8, 0.1% (wt/vol) SDS and 150 mM NaCl), then 10 min washing with 0.1x TE. After the 0.1x TE wash, the beads were resuspended in 100 µL of 0.1x TE and 10 µL used for nested PCR. Nested PCR reaction contained: 5 µL of 10X polymerase buffer (NEB, supplied with Deepvent(-Exo) enzme), 4 µL MgSO$_4$ (100 mM), 1 µL of Deepvent(-Exo), 1 µL NTP mix (0.5 mM each final), 1 µL each nested primers (stock concentration 100 µM), and 10 µL of beads. Nested PCR was done in a thermocycle with the following amplification steps: 1) one step of total denaturation: 15 min at 95°C; 2) 15 steps of amplification: 30 s at 95°C 30 s at 60°C, 5 min at 72°C; and 3) one step of extension: 5 min at 72°C. Nested reactions were diluted 50-fold in 0.1x TE, and serial dilutions were prepared to determine linear range of amplification. Fold enrichment for a given locus was calculated using the comparative Ct method, and then normalizing the samples to the measurements of the wild-type results.

## Comet assay

U2OS cells were grown in DMEM/10% FBS media in the presence of 1 µg/mL doxycyclin in 6-well plates at a very sparse seeding density. After 3 days, the cells were treated with DNA damaging agents, harvested by trypsinization, and rinsed in 1 mL cold PBS. Olive moments of damaged DNA were measured using OxiSelect Comet Assay Kit (3-Well Slides) (Cellbiolabs, #STA-350).

## Acknowledgements

Work in the TTP laboratory is supported in part by the Cancer Prevention and Research Institute of Texas (RP160667). The KMM laboratory is supported by the NIH National Cancer Institute (R01CA198279 and RO1CA201268) and the American Cancer Society (RSG-16-042-01-DMC). The work of Justin Leung is supported by the NIH National Cancer Institute (K22CA204354). We thank Steve Hanes, Nicholas Proudfoot, Jeff Corden, John Petrini, Hannah Klein, Lorraine Symington, Jim Haber, Michael Lieber, Steve Jackson, Patrick Calsou, Eric Campeau, and Paul Kaufman for reagents including yeast strains and plasmids.

## Additional information

### Funding

| Funder | Grant reference number | Author |
| --- | --- | --- |
| Cancer Prevention and Research Institute of Texas | RP160667 | Nodar Makharashvili<br>Yizhi Yin |
| Howard Hughes Medical Institute | | Sucheta Arora<br>Qiong Fu<br>Xuemei Wen<br>Ji-Hoon Lee<br>Chung-Hsuan Kao<br>Tanya T Paull |
| National Cancer Institute | K22CA204354 | Justin WC Leung |
| National Cancer Institute | R01CA198279 | Kyle M Miller |
| National Cancer Institute | RO1CA201268 | Kyle M Miller |
| American Cancer Society | RSG-16-042-01-DMC | Kyle M Miller |
| Cancer Prevention and Research Institute of Texas | RP160667 | Tanya T Paull |

The funders had no role in study design, data collection and interpretation, or the decision to submit the work for publication.

### Author contributions

Nodar Makharashvili, Conceptualization, Data curation, Formal analysis, Validation, Investigation, Visualization, Methodology, Writing—original draft, Writing—review and editing; Sucheta Arora, Yizhi Yin, Xuemei Wen, Ji-Hoon Lee, Chung-Hsuan Kao, Justin WC Leung, Investigation, Methodology, Writing—review and editing; Qiong Fu, Conceptualization, Investigation, Methodology, Writing—review and editing; Kyle M Miller, Supervision, Validation, Writing—review and editing; Tanya T Paull, Conceptualization, Resources, Formal analysis, Supervision, Funding acquisition, Investigation, Writing—original draft, Project administration, Writing—review and editing

### Author ORCIDs

Ji-Hoon Lee https://orcid.org/0000-0001-7387-935X
Tanya T Paull http://orcid.org/0000-0002-2991-651X

### Decision letter and Author response

Decision letter https://doi.org/10.7554/eLife.42733.030
Author response https://doi.org/10.7554/eLife.42733.031

## Additional files

### Supplementary files

• Supplementary file 1. Yeast strain list.
DOI: https://doi.org/10.7554/eLife.42733.018
• Supplementary file 2. Sae2 ChIP-seq data.

DOI: https://doi.org/10.7554/eLife.42733.019
• Supplementary file 3. CtIP RNA-seq data.
DOI: https://doi.org/10.7554/eLife.42733.020
• Supplementary file 4. Oligonucleotides.
DOI: https://doi.org/10.7554/eLife.42733.021
• Transparent reporting form
DOI: https://doi.org/10.7554/eLife.42733.022

### Data availability

All data generated or analyzed during this study are included in the manuscript and supporting files. Sequencing data has been deposited in GEO (accession number GSE122782).

The following dataset was generated:

| Author(s) | Year | Dataset title | Dataset URL | Database and Identifier |
|-----------|------|---------------|-------------|-------------------------|
| Tanya T Paull | 2018 | Sequencing data from Sae2/CtIP prevents R-loop accumulation in eukaryotic cells | https://www.ncbi.nlm.nih.gov/geo/query/acc.cgi?acc=GSE122782 | NCBI Gene Expression Omnibus, GSE122782 |

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
