## [Decision Letter]

[Editors’ note: a previous version of this study was rejected after peer review, but the authors submitted for reconsideration. The first decision letter after peer review is shown below.]

Thank you for submitting your work entitled "Sae2/CtIP prevents R-loop accumulation in eukaryotic cells" for consideration by *eLife*. Your article has been reviewed by three peer reviewers, one of whom is a member of our Board of Reviewing Editors, and the evaluation has been overseen by a Reviewing Editor and a Senior Editor. The reviewers have opted to remain anonymous.

Our decision has been reached after consultation between the reviewers. Based on these discussions and the individual reviews below, we regret to inform you that your work will not be considered further for publication in *eLife* at this moment. We believe that this is very interesting work with exciting results, but it needs further analysis to support the model. However, we would be happy to consider a resubmission of your work, if you can appropriately respond to the reviewers.

As I said, this is an interesting study performed in yeast and human cells to show that CtIP is involved in resolving R-loops in conjunction with XPG. By first observing the ability of Sen1 DNA-RNA helicase, involved in transcription termination, to suppress some of the phenotypes of *sae2*∆ mutants such as CPT sensitivity and R-loop accumulation in the rDNA, authors conclude that yeast Sae2 is important for R-loop removal. They go on to show that Sae2 is found at sites of high transcription, especially upon treatment with CPT in S phase, and that its deletion leads to transcription-dependent damage sensitivity and Pol II stalling following CPT. Notably, most of these findings also apply for CtIP function in human cells. CtIP depletion triggers sensitivity to CPT that is rescued by Senataxin (the Sen1 human ortholog) overexpression. They find that RNA-DNA hybrids accumulate upon CTIP-loss or XPG-loss at laser-induced damage, based on binding of an mCherry-RNaseH sensor. Moreover, CtIP depletion decreases ssDNA break, suggesting that, as XPG, CtIP contributes to R-loop dissolution by generating a ssDNA break promoting the activity of senataxin and other hybrids helicases. The authors show that CtIP and XPG reduce breaks formed by CPT using both comet assays and LM-PCR. The most puzzling data came from combining both XPG and CtIP depletion, since, although their co-depletion completely abolishes the generation of an ssDNA break, no R-loop accumulate in these conditions. The authors interpret these data, as a lack of R-loop extension in these conditions, that would make them undetectable. Together, these studies lead the authors to propose that XPG and CtIP work in a redundant manner to process R-loops by generating a break at the R-loop site and allowing access of SETX to the hybrids for hybrid resolution. They propose a model by which CtIP would promote cleavage of Flap structures as a step to resolve R-loops.

The work is very interesting and exciting and provides an intriguing model and important function of CTIP to understand. However, additional results are needed to test the model and rule out other possibilities. Several experiments require additional control; all studies with S9.6 should provide the control with RNaseH treatment in vitro to show that the signal is removed (S9.6 also recognizes dsRNA and data must assure that the signal detected corresponds to RNA-DNA hybrid). Although authors use Sen1 to remove R-loops, the data suggest that the terminator function of Sen1 has a role in the suppression of the CPT sensitivity and by extension the R-loops. In addition, the yeast data do not clearly support that RNAPII transcription is the major cause of R-loops. The authors are unable to show hybrids in RNAPII genes, but only in rDNA regions. Of particular note, the authors did not sufficiently establish the proposed order of events which place CtIP and XPG upstream of SETX. Is CtIP or XPG needed for Sen1/SETX to bind the DNA at the sites of hybrid formation, tested using the RNaseH sensor and ChIP assays? This would be one prediction of the model if nuclease processing is needed to generate access for Sen1/SETX. Could indeed the authors rule out that SETX and CtIP are not simply acting to reduce the same hybrid structures (albeit in different ways)? In summary, as it stands the authors need more data to substantiate their conclusions.

Reviewer #1:

This is an interesting study performed in yeast and human cells to show that CtIP is involved in resolving R-loops. By first observing the ability of Sen1 DNA-RNA helicase to suppress some of the phenotypes of *sae2∆* mutants such as CPT sensitivity and R-loops detected by a specific antibody, authors conclude that yeast Sae2 is important for R-loop removal. The study is extended to human cells, in which using a parallel approach the authors conclude that CtIP have similar role. In this case, they use laser micro-irradiation and siXPG cells, which allow the authors to propose a model by which CtIP would be able to cleave Flap structures as a step to resolve R-loops. This is a new and a priori unexpected involvement of a DSB processing factor in R-loop removal that merits being considered for publication. However, the study has a number of issues that need to be resolved before it can be published. There are three important points that need to be solved first, apart from the essential experiments suggested below.

- All studies with S9.6, except one in Figure 6A, did not provide the control with RNase H1 to show that the signal is removed. It is known that S9.6 also recognizes dsRNA and it is not possible to make conclusions with this antibody without removing the signal with RNase H1.

- Although authors use Sen1 to remove R-loops, the data suggest that the terminator function of Sen1 has a role in the suppression of the CPT sensitivity and by extension the R-loops. It is necessary to show that RNH1 overexpression, not just Sen1/SETX also suppresses the CPT phenotype in yeast and that of micro-irradiated human cells.

- Authors propose that CtIP/Sae2 may have a new FLAP endonucleolytic function similar to XPG. This conclusion is based on the similar phenotypes observed for CtIP and XPG-depleted cells. However, a similar phenotype may be produced by different mechanisms, and R-loop dynamics may respond to many different factors. The in vivo data showing that in the double depletion in human cells they have equal amount of R-loops than the simple is inconsistent with a model implying that an ssDNA nick (as that caused by S1 in vitro experiment) stabilizes R-loops. The in vivo data do not seem to support the model proposed since if CtIP is not able to cleaves the FLAP structure, R-loops would not be stabilized. In this context, the S1 experiment is really confusing and unnecessary in this study (the Lieber lab already showed in the past that a nick in the DNA increases the accumulation of R-loops).

- Provided the know function of CtIP in DSB resection and the relevance of transcription-replication collisions as a way to produce stalled and collapsed forks and considering the possibility that it could also lead to fork reversal generating a one-ended DSB-like structure susceptible of being resected by CtIP to restart the fork, it may be possible that the inability of CtIP to act at broken or reversal replication forks stalled in front of an R-loop lead to an accumulation of these hybrids as structures causing the lethality in CPT or microirradiation and Sae2/CtIP cells. Consequently, it is necessary to show that in vitro CtIP has the suggested activity. The assay done with S1 should indeed be performed with CtIP.

In Figure 1, authors should show overexpression of RNaseH to demonstrate that R-loops are the cause of CPT sensitivity in *sae2∆* strain.

Since the CPT sensitivity phenotype is observed in S-phase, it would be important to address if Sen1 and RNaseH1 rescue *sae2∆* sensitivity to HU. Could it be transcription/replication collisions not R-loop dependent?

A DRIP-qPCR analysis of some of the genes with increased Sae2 recruitment in the ChIP-seq to validate the data.

In Figure 1E, the control DRIP samples with RNaseH1 treatment is not shown.

No statistical analysis is supplied for any of the yeast tables. This is essential in Figure 3A, where authors claim that o/e SEN1 reduces RNA Pol II accumulation. Again, an RNaseH1 o/e should be included.

In Figure 4A, authors should show overexpression of RNaseH as a way to demonstrate that R-loops are the cause of CPT sensitivity in cells depleted of CtIP.

Immunofluorescences in Figure 5C should have RNase H -treated cells as control. Image contrast could be improved and also show the channels separately.

How do authors explain the reduction on R-loops compare to siC levels when both nucleases CtIP and XPG are missing? This is odd.

Regarding the ssDNA breaks assay, due to the CtIP role in resection it is expected to have less breaks after DNA damage. Authors could perform the assay in a background known to increase R-loops, such as sen1 or rnh1 rnh2 and see if CtIP depletion reduces the R-loop-dependent DNA breaks.

The yeast data do not seem to support that RNAPII transcription is the major cause of R-loops. The authors are unable to show hybrids in RNAPII genes, but only in rDNA regions. Indeed, it makes sense from the data that CPT, that it is the rDNA region, in which the Tollervey lab has shown the relevance of Top1 to prevent R-loop formation, where the major intermediate leading to lethality may accumulate after CPT. Is there any data showing an involvement of Sen1 in RNAPI transcription termination? This needs to be considered, since at the end it is not really clear which function of Sen1 is able to suppress the sensitivity to CPT of *sae2∆* cells.

The enrichment of Sae2 at 176 sites of transcription in S phase may not necessarily related with R-loops. Authors do not show whether such enrichment is reduced by RNase H overexpression. At least should be done by DRIP-qPCR to validate the data using RNase H.

In the genome-wide analysis, the fact that RNAPII is found at high levels at sites previously described to from R-loops does not implies that R-loops are accumulated in *sae2∆* strains. It is required to reduce the signal by overexpressing RNH1. Overexpression of Sen1 produces a reduction in signal, but this may be due to the terminator function of Sen1. For this reason, authors need to revert the phenotype with a protein not involved in termination, but just in R-loop removal, such as RNH1.

I am not sure what the results of Figure 4B really says. If CPT causes breaks, it should not be surprising that CtIP increases the sensitivity. Provided the impact of transcription in collisions or increase in damage by other ways, the action of DRB should not be surprising. However, the reduction of the sensitivity phenotype is very small. Results could be repeated using another inhibitor of transcription such as α-amanitin.

In Figure 5 it is necessary to show that all signals are reduced by RNase H.

*Reviewer #2:*

In this manuscript, Makharashvili et al., report a novel function for Sae2/CtIP in both yeast and human cells: They propose that this endonuclease, known to promote resection during DSB repair, also contributes in R-loops (DNA-RNA hybrids) dissolution, in conjunction with XPG. First, they found that yeast sen1, an RNA-DNA helicase involved in transcription termination, as well as transcription inhibitors, rescue the CPT sensitivity of sae2 deficient strains. They further report that Sae2 binds to highly transcribed genes following exposure to CPT in S phase, that it correlates with accumulation (pausing) of RNA Polymerase II at these positions in CPT treated cells, and that R-loops accumulate in Sae2 deficient strain (at the rDNA locus). Notably, most of these findings also apply for CtIP function in human cells: CtIP depletion triggers sensitivity to CPT that is rescued by Senataxin (the Sen1 human ortholog) overexpression and R-loops accumulate in CtIP-depleted cells, as they do in XPG-depleted cells, as previously reported. Moreover, CtIP depletion decreases ssDNA break, suggesting that, as XPG, CtIP contributes to R-loop dissolution by generating a ssDNA break promoting the activity of senataxin and other hybrids helicases. All these data identify Sae2/CtIP as a key regulator of R-loops stability on yeast and human genome. The most puzzling data came from combining both XPG and CtIP depletion, since, although their co depletion completely abolishes the generation of an ssDNA break, no R-loop accumulate in these conditions. The authors interpret these data, as a lack of R-loop extension in these conditions, that would make them undetectable.

Given the threat of R-loop on genome stability that has been clearly described recently, identifying the mechanisms that function to regulate their occurrence on the genome is of prime interest. Additionally, the model presented here is very interesting and exciting. Yet I feel that (i) the model would be more convincing if the same loci were analyzed across the Figures, (see main point 1 below) and (ii) some key experiment are lacking to fully support the model (see main point 2).

For the yeast part, the authors need to show that indeed, R-loops accumulates at sites that recruit sae2 upon CPT treatment. They show Figure 2E that they do accumulate on rDNA, but to be consistent with Figure 2C, and Figure 3A-B and to demonstrate convincingly the relationship at specific loci between Sae2 and R-loops, the authors need to report the accumulation of R-loops at ENO1, ADH2, FBA1, as well as SNR13, SNR5. Additionally, the authors need to extend their Sae2-flag ChIP-seq data analysis more specifically by comparing their data with DRIP-seq yeast genome profiles (see my detailed points on Figure 2).

For the mammalian cells part, the fact that codepletion of XPG and CtIP shows no R-loops accumulation but deficient ssDNA breakage is really puzzling. The authors propose that the hybrids still accumulate in these conditions but are not detectable using their assay (S9.6 and mutated cherry RNAseH). The author needs to demonstrate that this is the case. Indeed, that CtIP depletion rescues R-loop accumulation in XPG-depleted cells may also indicate that CtIP lies downstream XPG mediated-ssDNA cleavage, hence challenging their model. I realize this may be difficult to assess, but this is an important piece of data that is lacking to support their current model; maybe the authors could try at one locus non denaturing sodium bisulfite treatment to detect ssDNA (as in Ginno et al., 2012 or Loomis et al., 2014)?

Figure 1: Sen 1 also rescues viability upon CPT in mre 11 deficient strain. Can the author comment and integrate the MRX complex in their R-loop processing model?

Figure 2: The ChIP-seq data should be more described, and a full supplemental Figure should be provided. Do the 45 peaks also overlap with the 176 peaks in presence of CPT? If yes, can we see some examples on browser? Can the author show a scatter plot where for each gene, the transcription level is plotted against the CPT-treated enrichment of Sae2 (normalized against gene length)? The author could also retrieve RNA Pol II mapping and perform the same analysis. Another very common way to show this type of data, would be to separate the yeast genes based on their expression level and show that Sae2 binding is statistically higher for the highly expressed genes subset compared to the low expressed set of genes. Finally, Sae2 binding in S phase upon CPT should be compared to DRIP-seq data from the Koshland lab (Wahba et al., 2016). Given that the authors show R-loop accumulation at rDNA, can they also report the change of Sae2 recruitment upon CPT on rDNA from their Sae2 ChIP-seq? Can we also see the snapshot for PDC1, SNR13, SNR5 for Sae2 distribution in CPT treated samples?

Figure 3: "We did not observe CPT induced RNA Pol II pausing at PDC1". It seems from the Figure that the author can actually see some Rbp2 increase (Figure 3B).

Figure 5: In their expression study, is there some overlap between genes modified following CPT treatment and genes modified after CtIP depletion? Can the author directly show the DRIP-seq count on the different gene subsets (WT -vs +CPT, WTvsCtiP -CPT and WT vs CtIP + CPT) compared to random genes (together with p values)?

Figure 6: Part of the Figure 6B, can the author show the R-loop level by DRIP qPCR (only at the b-actin gene) upon XPG and CtIP depletion?

Figure 8: the assay presented Figure 8D is very nice. Could the author also perform XPG only and CtIP/XPG depletion to further validate that indeed ssDNA breakage is absent in these conditions?

Reviewer #3:

In this manuscript, the role of Sae2/CTIP in R-loop accumulation and processing is explored both in yeast and mammalian cells. Motivated by understanding the unexpected sensitivity of Sae2/CTIP-deficient cells to CPT-induced 3'-ssDNA lesions, Paull and colleagues explored the role of transcriptional regulation in survival of Sae2-deficient cells. Upon finding that SEN1 expression rescues this sensitivity and that its loss enhances sensitivity, they go on to show that Sae2 is found at sites of high transcription, especially upon treatment with CPT, and that its deletion leads to transcription-dependent damage sensitivity and Pol II stalling following CPT. Similarly, they find in mammalian cells that sensitivity due to CTIP loss can be rescued by SETX expression and that RNA-DNA hybrids accumulate upon CTIP-loss or XPG-loss at laser-induced damage, based on binding of an mCherry-RNaseH sensor. Hybrids are also elevated in the absence of damage and these nucleases both at genes know to form R-loops and R-loop inducible sites. Finally, the authors show that CtIP and XPG reduce breaks formed by CPT using both comet assays and LM-PCR. Together, these studies lead the authors to propose that XPG and CtIP work in a redundant manner to process R-loops by generating a break at the R-loop site and allowing access of SETX to the hybrids for hybrid resolution.

While this is an intriguing model and important function of CTIP to understand, I think that the model is somewhat premature at this point and additional results are needed to test the model and rule out other possibilities. Several experiments require additional controls as well. Of particular note, I don't think the authors have sufficiently established the proposed order of events which place CtIP and XPG upstream of SETX. Is CtIP or XPG needed for Sen1/SETX to bind the DNA at the sites of hybrid formation, tested using the RNaseH sensor and ChIP assays? This would be one prediction of the model if nuclease processing is needed to generate access for Sen1/SETX. Also, can the authors rule out that SETX and CtIP are not simply acting to reduce the same hybrid structures (albeit in different ways)? As it stands the authors need more data to substantiate their conclusions.

Where does the CPT-induced break fit into the authors' model? The break induced by CtIP/XPG would be in addition to this CPT-induced break. This is omitted from Figures and models and a somewhat confusing point.

Figure 2: As a specificity control the authors should test whether Sae2 binding by ChIP is blocked by transcription inhibition.

Figure 2: Since the DRIP did not work well in yeast, it would be helpful to show that Sae2 binding by ChIP is rescued by overexpression of RNaseH or Sen1 in yeast cells. Similarly, they could ask if RNaseH overexpression rescues CPT effects on survival.

Figure 3: The authors need to analyze several other genes that are not expressed or expressed at lower levels to show specificity of the effect of CtIP loss at highly expressed genes. Right now they examine two genes, PDC1 and NRD1 as negative controls (if I understand this correctly) and Pol II binding is actually elevated at the first site.

Figure 4: Does RNaseH overexpression rescue viability of CTIP deletion? Also, please comment why in Figure 4 hybrids increase with DRB treatment in WT cells?

Figure 4, Figure 5 and Figure 6: Is the nuclease activity of XPG required for any of the observed effects? Testing this in at least one of the assays would be helpful.

Figure 5D: Does RNaseH expression in vivo and RNaseH treatment in vitro reduce the increased hybrid signal (which is actually very weak). This is an important control.

Figure 5E/F: What is the rationale for both increases and decreases in transcription overlapping with site of hybrid formation? I don't understand the implications of the RNA-seq data. Also, Figure 5F is missing in the Figure, making these data difficult to evaluate.

Figure 5: What is the impact of CtIP and XPG loss, or loss of both, on the cell cycle. Are breaks or hybrid levels reduced due to alterations in cell cycle progression (e.g. failure to enter S phase) or accumulation in G2. It seems in WT cells hybrids are naturally higher in G2 and if breaks depend on S phase entry changes in cell cycle could affect the interpretation.

Figure 6: Why are there no more breaks upon addition of CPT? Shouldn't the CPT itself lead to break formation even without the action of XPG and CtIP? Also, alkaline comet is not specific for SSB detection – these could be DSBs. I suggest the authors use caution in the way they present these data.

The idea that the R-loops are too small to be detected when both XPG and CtIP are lost is not well supported. Isn't it possible that there is no R-loop under these conditions and that RNAPII has simply stalled and is the toxic lesion? Can RNaseH expression rescue sensitivity due to loss of both XPG and CtIP? That is one test of this idea. Or can evidence for a smaller R-loops be obtained using bisulfite sequencing. Minimally I think other models should be considered.

[Editors’ note: what now follows is the decision letter after the authors submitted for further consideration.]

Thank you for submitting your article "Sae2/CtIP prevents R-loop accumulation in eukaryotic cells" for consideration by *eLife*. Your article has been reviewed by the same three peer reviewers that revised the first submission, and the evaluation has been overseen by the same Reviewing Editor and Jessica Tyler as the Senior Editor. The reviewers have opted to remain anonymous.

The reviewers have discussed the reviews with one another and the Reviewing Editor has drafted this decision to help you prepare a revised submission.

Summary:

Makharashvili et al., report a novel function for Sae2/CtIP in both yeast and human cells, and propose that Sae2/CtIP, known to promote resection during DSB repair, also contributes in R-loop (DNA-RNA hybrids) dissolution, in conjunction with XPG. In this revised version, the authors have addressed to a reasonable extent my main points. More specifically they are now consistent throughout the manuscript, and report Sae2 binding and R-loop accumulation in Sae2 deleted strain at the same loci. They have also now changed their discussion to take more hypotheses into account. They have also largely controlled their data by performing RNAseH treatment to validate their results obtained with S9.6 antibody. This manuscript is a significantly improved version of the previous one. Results are now clearer and convincing. However, the key conclusion of their model that CtIP cleaves the loop needs further support and clarification before acceptance.

Essential revisions:

The results on which the model proposed rely on CPT-treated cells (i.e. Figure 2A-E, Figure 4A-C, Figure 5F and Figure 8) (CPT creates ssDNA breaks with Top1-cc) or cells upon laser-induced breaks (i.e. Figure 4D-H). So, the experimental conditions used implies to start with a genotoxic-induced break. It is still therefore unclear why CtIP would be required for an additional break. Indeed, authors were not able to show the proposed activity of CtIP in vitro. Authors would need to show the action of CtIP in untreated cells (no CPT, no laser irradiation), with no break, to support their model that demands that CtIP cleaves the loop. The known action of the Sae2/CtIP nuclease activity on DNA resection as a requirement to remove the RNA-DNA hybrid accumulated at a break would make much more sense and fit better with the combined effect with XPG. Therefore, either authors provide new data with untreated cells to support their model as such or alternatively include in their model the breaks caused by CPT/laser irradiation to explain the role of CtIP under such conditions.

---

## [Author Response]

[Editors’ note: the author responses to the first round of peer review follow.]

As I said, this is an interesting study performed in yeast and human cells to show that CtIP is involved in resolving R-loops in conjunction with XPG. By first observing the ability of Sen1 DNA-RNA helicase, involved in transcription termination, to suppress some of the phenotypes of sae2∆ mutants such as CPT sensitivity and R-loop accumulation in the rDNA, authors conclude that yeast Sae2 is important for R-loop removal. They go on to show that Sae2 is found at sites of high transcription, especially upon treatment with CPT in S phase, and that its deletion leads to transcription-dependent damage sensitivity and Pol II stalling following CPT. Notably, most of these findings also apply for CtIP function in human cells. CtIP depletion triggers sensitivity to CPT that is rescued by Senataxin (the Sen1 human ortholog) overexpression. They find that RNA-DNA hybrids accumulate upon CTIP-loss or XPG-loss at laser-induced damage, based on binding of an mCherry-RNaseH sensor. Moreover, CtIP depletion decreases ssDNA break, suggesting that, as XPG, CtIP contributes to R-loop dissolution by generating a ssDNA break promoting the activity of senataxin and other hybrids helicases. The authors show that CtIP and XPG reduce breaks formed by CPT using both comet assays and LM-PCR. The most puzzling data came from combining both XPG and CtIP depletion, since, although their co-depletion completely abolishes the generation of an ssDNA break, no R-loop accumulate in these conditions. The authors interpret these data, as a lack of R-loop extension in these conditions, that would make them undetectable. Together, these studies lead the authors to propose that XPG and CtIP work in a redundant manner to process R-loops by generating a break at the R-loop site and allowing access of SETX to the hybrids for hybrid resolution. They propose a model by which CtIP would promote cleavage of Flap structures as a step to resolve R-loops.The work is very interesting and exciting and provides an intriguing model and important function of CTIP to understand. However, additional results are needed to test the model and rule out other possibilities. Several experiments require additional control; all studies with S9.6 should provide the control with RNaseH treatment in vitro to show that the signal is removed (S9.6 also recognizes dsRNA and data must assure that the signal detected corresponds to RNA-DNA hybrid). Although authors use Sen1 to remove R-loops, the data suggest that the terminator function of Sen1 has a role in the suppression of the CPT sensitivity and by extension the R-loops. In addition, the yeast data do not clearly support that RNAPII transcription is the major cause of R-loops. The authors are unable to show hybrids in RNAPII genes, but only in rDNA regions. Of particular note, the authors did not sufficiently establish the proposed order of events which place CtIP and XPG upstream of SETX. Is CtIP or XPG needed for Sen1/SETX to bind the DNA at the sites of hybrid formation, tested using the RNaseH sensor and ChIP assays? This would be one prediction of the model if nuclease processing is needed to generate access for Sen1/SETX. Could indeed the authors rule out that SETX and CtIP are not simply acting to reduce the same hybrid structures (albeit in different ways)? In summary, as it stands the authors need more data to substantiate their conclusions.

Thank you for the in-depth comments. From the summary we received, it seems that the major issues have to do with: (1) confirmation of the specificity of S9.6 immunoprecipitations, (2) demonstration of the ability of RNaseH suppression in addition to SEN1/Senataxin suppression of CtIP depletion phenotypes, (3) demonstration of R-loops at RNAPII genes in yeast, and (4) further investigation and validation of the ideas in our proposed model. We have performed several additional experiments to address these points, as discussed below. We think the manuscript is significantly improved with these revisions and additions to the data. We have also revised our thinking about the model and have incorporated some alternative scenarios into the discussion in the main text. We address these four main questions first and then answer the reviewer questions below.

1) RNaseH in vitro treatment for validation of S9.6 IPs:

We repeated the S9.6 DRIP-qPCR experiment using pre-treatment with RNaseH in vitro as a control. These results, shown in Figure 6C, and Figure 6—Figure supplement 3 that the S9.6 signal is in fact due to RNA-DNA hybrids as it is eliminated by RNaseH.

2) Another suggestion of the reviewers was to show RNaseH suppression of phenotypes in yeast and in human cells, which would confirm the R-loop-related role of SEN1/SETX.

Galactose-induced overexpression of RNH1 also suppresses the sensitivity of *Δsae2* yeast cells to camptothecin and to MMS, as shown in Figure 1.

We also performed this experiment in human cells by overexpressing RNaseH in cells depleted of CtIP. We found that this treatment also reduces R-loops in CtIP-depleted cells, as measured by DRIP-qPCR. This result is now shown in Figure 6B and D.

3) Is RNAPII transcription the cause of R-loop accumulation in *sae2∆* yeast cells? To answer this question we re-visited DRIP experiments in yeast. We observed a statistically significant increase in R-loops at highly transcribed loci in *sae2∆* cells, as in Figure 2E (ADH1, ENO2, FBA1).

4) To address whether the proposed model of SEN1/SETX recruitment to transcription sites via Sae2/CtIP is correct.

We decided to determine if SEN1 and SETX recruitment to relevant sites in both yeast and human cells is Sae2/CtIP-dependent. In yeast, we used an HTB-tagged SEN1 strain and examined recruitment to several of the loci where we observed high levels of Sae2 protein and also R-loop accumulation. This experiment shows that SEN1 is actually present at higher levels than in wild-type strains (these isolations were all done in S phase with CPT treatment) (Figure 3C).

In human cells we performed ChIP-qPCR experiments to determine if endogenous Senataxin is present at sites of R-loop accumulation. We used our inducible Sγ3 locus for these experiments since we are able to control the transcription at this site. Similar to the experiment in yeast, we found that Senataxin occupancy is actually higher in CtIP-depleted cells than in wild-type, and that this is dependent on active transcription (Figure 6H). It was also higher with XPG depletion but the p value for this comparison was 0.06.

We conclude from this that the recruitment of SEN1/SETX protein is not impaired, yet there seems to be a significant dysfunction of the enzyme since levels of R-loops are high and RNAPII is found to be stalled at these sites. One possibility is that SEN1/SETX recruitment to the site is mediated by association with the polymerase or other factors, yet the enzyme is not able to access the R-loop efficiently because of a problem with the structure of the DNA at the locus (our initial model). It is also possible that Sae2/CtIP is in a separate, parallel pathway to SEN1/SETX and that SEN1 overexpression (or RNaseH overexpression) helps in *sae2∆* or CtIP-depleted cells because this alternate pathway is augmented by the increased removal of RNA-DNA hybrids. We now present both of these models in the main text and in Figure 7.

Reviewer #1:This is an interesting study performed in yeast and human cells to show that CtIP is involved in resolving R-loops. By first observing the ability of Sen1 DNA-RNA helicase to suppress some of the phenotypes of sae2∆ mutants such as CPT sensitivity and R-loops detected by a specific antibody, authors conclude that yeast Sae2 is important for R-loop removal. The study is extended to human cells, in which using a parallel approach the authors conclude that CtIP have similar role. In this case, they use laser micro-irradiation and siXPG cells, which allow the authors to propose a model by which CtIP would be able to cleave Flap structures as a step to resolve R-loops. This is a new and a priori unexpected involvement of a DSB processing factor in R-loop removal that merits being considered for publication. However, the study has a number of issues that need to be resolved before it can be published. There are three important points that need to be solved first, apart from the essential experiments suggested below.1) All studies with S9.6, except one in Figure 6A, did not provide the control with RNase H1 to show that the signal is removed. It is known that S9.6 also recognizes dsRNA and it is not possible to make conclusions with this antibody without removing the signal with RNase H1.

We have repeated the DRIP-qPCR experiments and performed RNaseH treatment of the samples before the IP (shown in Figure 6A and Figure 6—Figure supplement 3), showing that the signal is specific to RNA-DNA hybrids.

- Although authors use Sen1 to remove R-loops, the data suggest that the terminator function of Sen1 has a role in the suppression of the CPT sensitivity and by extension the R-loops. It is necessary to show that RNH1 overexpression, not just Sen1/SETX also suppresses the CPT phenotype in yeast and that of micro-irradiated human cells.

See Figure 1 and Figure 6B,D.

- Authors propose that CtIP/Sae2 may have a new FLAP endonucleolytic function similar to XPG. This conclusion is based on the similar phenotypes observed for CtIP and XPG-depleted cells. However, a similar phenotype may be produced by different mechanisms, and R-loop dynamics may respond to many different factors. The in vivo data showing that in the double depletion in human cells they have equal amount of R-loops than the simple is inconsistent with a model implying that an ssDNA nick (as that caused by S1 in vitro experiment) stabilizes R-loops. The in vivo data do not seem to support the model proposed since if CtIP is not able to cleaves the FLAP structure, R-loops would not be stabilized. In this context, the S1 experiment is really confusing and unnecessary in this study (the Lieber lab already showed in the past that a nick in the DNA increases the accumulation of R-loops).

We find that the double depletion of both CtIP and XPG generates a very low level of R-loops, similar to wild-type cells, whereas each single depletion shows high R-loops. So, the double depletion is not equivalent to either of the single depletions. To explain this, we propose that extensive R-loops are not being formed in the double depletion cells because of a lack of single-strand processing. The Lieber laboratory showed previously that a nick in either strand strongly promotes stabilization of R-loops, so our hypothesis is consistent with this idea. We agree that we don't need to show the nick inducing R-loops since this was published previously.

The model is simply a diagrammatic set of working hypotheses that we are using to frame questions around this data; we are certainly not claiming to have proven everything in the model, and the discussion is modified now to better reflect this view. We know that CtIP and XPG are not completely redundant because we see obvious R-loop accumulation in the absence of either factor. We are working toward understanding the differences between their activities using purified components, but this is beyond the scope of the work being considered here.

- Provided the know function of CtIP in DSB resection and the relevance of transcription-replication collisions as a way to produce stalled and collapsed forks and considering the possibility that it could also lead to fork reversal generating a one-ended DSB-like structure susceptible of being resected by CtIP to restart the fork, it may be possible that the inability of CtIP to act at broken or reversal replication forks stalled in front of an R-loop lead to an accumulation of these hybrids as structures causing the lethality in CPT or microirradiation and Sae2/CtIP cells. Consequently, it is necessary to show that in vitro CtIP has the suggested activity. The assay done with S1 should indeed be performed with CtIP.

We have made R-loops in vitro using prokaryotic and phage RNA polymerases but have not observed significant levels of processing of these structures. Considering the known association between CtIP and human transcription-associated complexes though, perhaps it is not surprising that a reconstitution with heterologous polymerase would not work here.

It is certainly possible that some of the consequences of CtIP depletion involve fork reversal. We now have alleles of CtIP that can separate the function of CtIP in canonical DSB processing from the nuclease-related function, so we are in the process of testing these.

In Figure 1, authors should show overexpression of RNaseH to demonstrate that R-loops are the cause of CPT sensitivity in sae2∆ strain.

As shown in Figure 1, we do observe suppression of CPT and MMS sensitivity by RNH1 in a *∆sae2* strain. We also show that RNaseH overexpression in human U2OS cells reduces R-loops in CtIP-depleted cells (Figure 6B,D).

Since the CPT sensitivity phenotype is observed in S-phase, it would be important to address if Sen1 and RNaseH1 rescue sae2∆ sensitivity to HU. Could it be transcription/replication collisions not R-loop dependent?

We tested the HU sensitivity of *sae2* strains to hydroxyurea with either *RNH1* or *SEN1* overexpression and found no evidence for suppression of sensitivity under these conditions (Author response image 1). Stalling of replication forks, although toxic in a *sae2* background, does not appear to be related to R-loop removal.

**Author response image 1. respfig1:** Overexpression of *RNH1* or SEN1 does not suppress the sensitivity of *Δsae2* strains to hydroxyurea (HU) exposure. (**A**). *RNH1* was induced from the GAL1 promoter on a CEN plasmid in *Δsae2* compared to *Δsae2* with vector only. Viability with or without HU (100 mM) was assessed at 48 hours or 72 hours as indicated. All plates contain galactose. (**B**). SEN1 was overexpressed from a CEN plasmid in*Δsae2* compared to *Δsae2* with vector only. Viability with or without CPT (5 μg/ml) was assessed at 48 or 72 hours as indicated on glucose-containing media.

A DRIP-qPCR analysis of some of the genes with increased Sae2 recruitment in the ChIP-seq to validate the data.

See point #3 above, and Figure 2E.

In Figure 1E, the control DRIP samples with RNaseH1 treatment is not shown.

We show in vitro treatment of samples with RNaseH in Figure 2E as well as in Figure 6A and Figure 6—Figure supplement 3. Figure 1 doesn't contain any DRIP experiments.

No statistical analysis is supplied for any of the yeast tables. This is essential in Figure 3A, where authors claim that o/e SEN1 reduces RNA Pol II accumulation. Again, an RNaseH1 o/e should be included.

p values have been added to Figure 3. We do not have RNaseH1 overexpression in this case but the fact that RNH1 overexpression rescues growth of *Δsae2* cells to CPT and MMS (Figure 1) shows that the role of RNA removal is critical.

In Figure 4A, authors should show overexpression of RNaseH as a way to demonstrate that R-loops are the cause of CPT sensitivity in cells depleted of CtIP.

We now show this in Figure 1 in yeast as well as in Figure 6 in human cells.

Immunofluorescences in Figure 5C should have RNase H -treated cells as control. Image contrast could be improved and also show the channels separately.

We expressed RNaseH in cells with CtIP depletion, as shown in Author response image 2 and in Figure 5. Expression of RNaseH reduced levels of RNA-DNA hybrids recognized by S9.6.

**Author response image 2. respfig2:** CtIP, XPG, or Senataxin-depleted cells exhibit higher levels of RNA-DNA hybrids. (**A**) Human U2OS cells expressing shRNA specific for CtIP with or without RNaseH expression as indicated were analyzed using immunofluorescence with S9.6 antibody and with antinucleolin antibody which stains the nucleoli. S9.6 signal overlapping with nucleolin signal was substracted from the total signal and the data was normalized to the size of the nucleus. (**B**) Quantification of >50 cells from each cell line was performed. Error bars represent S.E.M. **** denotes p < 0.0001 using Student's two-tailed T test with comparisons as indicated.

How do authors explain the reduction on R-loops compare to siC levels when both nucleases CtIP and XPG are missing? This is odd.

The idea is that nicks in DNA promote the extension and stabilization of R-loops. This has been shown in vitro by Lieber and colleagues and we have confirmed this result (previously shown in one of the supplementary Figures). In the absence of a single-strand break, RNA-DNA hybrids are extremely constrained topologically, and the region of the RNA-DNA hybrid is likely to be inaccessible because of the presence of the non-template strand of DNA. So, we are proposing that, in the absence of strand breaks, there is a nascent structure that is not efficiently recognized by S9.6 or RNaseH but yet is toxic to cells and needs to be removed. We speculate that this could be a small R-loop or a paused polymerase associated with a small R-loop but do not know exactly what this structure is.

Regarding the ssDNA breaks assay, due to the CtIP role in resection it is expected to have less breaks after DNA damage. Authors could perform the assay in a background known to increase R-loops, such as sen1 or rnh1 rnh2 and see if CtIP depletion reduces the R-loop-dependent DNA breaks.

Actually, we would expect more breaks in the absence of CtIP if its role is primarily in double-strand break repair because presumably there would be more unresolved breaks. In the manuscript we are using an alkaline comet assay which measures single-strand breaks, and here we always observe fewer in the absence of CtIP. Resection of double-strand breaks would not be expected to change the total number of single-strand breaks. Unfortunately, we do not have any method to measure single-strand breaks in yeast as the reviewer suggests. In human cells we have measured single-strand breaks in XPG-depleted cells (which have higher R-loops), and in this context, removal of CtIP also lowers the number of single-strand breaks (Figure 8B).

The yeast data do not seem to support that RNAPII transcription is the major cause of R-loops. The authors are unable to show hybrids in RNAPII genes, but only in rDNA regions. Indeed, it makes sense from the data that CPT, that it is the rDNA region, in which the Tollervey lab has shown the relevance of Top1 to prevent R-loop formation, where the major intermediate leading to lethality may accumulate after CPT. Is there any data showing an involvement of Sen1 in RNAPI transcription termination? This needs to be considered, since at the end it is not really clear which function of Sen1 is able to suppress the sensitivity to CPT of sae2∆ cells.

See point #3 above. We do now show higher levels of R-loops at RNAPII genes in *sae2∆* strains with DNA damage. Sen1 is definitely involved in transcription termination, of non-coding RNAs as well as a subset of RNAPII-transcribed protein coding genes (*1*–*3*). We are not trying to eliminate termination as part of the possible mechanism here; in fact, we also show in Figure 1 the ability of PCF11 to partially suppress *sae2* DNA damage sensitivity, and PCF11 is primarily known as a transcription termination factor (*4*). PCF11 also promotes the association of SEN1 with RNAPII however, and both *sen1* and *pcf11* mutations in yeast induce higher R-loops (*4*). It is also important to point out that SEN1 promotes transcription termination by removal of nascent RNA made by RNA Pol I as well as by RNA Pol II (*5*), although we removed the rDNA data here is favor of the RNAPII data.

The enrichment of Sae2 at 176 sites of transcription in S phase may not necessarily related with R-loops. Authors do not show whether such enrichment is reduced by RNase H overexpression. At least should be done by DRIP-qPCR to validate the data using RNase H.

We now show that the sites we isolated with high Sae2 occupancy do have higher R-loops in *sae2∆* strains, and we show RNaseH treatment of the S9.6 IPs eliminates the signal. We also show that RNH1 expression reduces the sensitivity of *sae2∆* strains to both CPT and MMS, and we show that RNaseH overexpression in human cells reduces levels of R-loops in CtIP-depleted cells. We are pretty darn sure that these sites have R-loops. We are not saying that Sae2/CtIP is going to these sites because there are R-loops there, but that Sae2/CtIP goes to sites where high transcription is happening, and that in the absence of Sae2/CtIP, there are R-loops.

In the genome-wide analysis, the fact that RNAPII is found at high levels at sites previously described to from R-loops does not implies that R-loops are accumulated in sae2∆ strains. It is required to reduce the signal by overexpressing RNH1. Overexpression of Sen1 produces a reduction in signal, but this may be due to the terminator function of Sen1. For this reason, authors need to revert the phenotype with a protein not involved in termination, but just in R-loop removal, such as RNH1.

Overexpression of RNH1 does suppress the phenotype of *Δsae2* strains to both CPT and MMS (see Figure 1), and overexpression of wild-type RNaseH reduces levels of R-loops in CtIP-depleted human cells. Also, we found that the *sen1-R302W* allele rescues *Δsae2* DNA damage sensitivity similar to wild-type SEN1. This mutation has been reported to abrogate the terminator function of SEN1 by blocking interaction of SEN1 with Rpb1 (*6*), suggesting that the terminator function of SEN1 is not critical (Figure 1A). In contrast, mutation of the helicase domain does reduce the ability of SEN1 to suppress the phenotype, and a combination of this mutation with a *Δsae2* deletion generates extreme DNA damage sensitivity (Figure 1C).

I am not sure what the results of Figure 4B really says. If CPT causes breaks, it should not be surprising that CtIP increases the sensitivity. Provided the impact of transcription in collisions or increase in damage by other ways, the action of DRB should not be surprising. However, the reduction of the sensitivity phenotype is very small. Results could be repeated using another inhibitor of transcription such as α-amanitin.

The CPT sensitivity assay in Figure 4B shows partial suppression of the sensitivity observed in CtIP-depleted cells by DRB treatment. The fact that we see partial restoration is expected because the transcription-associated role of CtIP that we are describing in this work is only part of its diverse biological functions, which also include promoting the nuclease activity of Mre11 at double-strand break sites. We would not expect transcription inhibition to necessarily affect this aspect of its function if the double-strand break is the critical lesion. We also previously tested α-amanitin but we found it was too toxic in this type of assay.

In Figure 5 it is necessary to show that all signals are reduced by RNase H.

See point #1 above.

Reviewer #2:In this manuscript, Makharashvili et al., report a novel function for Sae2/CtIP in both yeast and human cells: They propose that this endonuclease, known to promote resection during DSB repair, also contributes in R-loops (DNA-RNA hybrids) dissolution, in conjunction with XPG. First, they found that yeast sen1, an RNA-DNA helicase involved in transcription termination, as well as transcription inhibitors, rescue the CPT sensitivity of sae2 deficient strains. They further report that Sae2 binds to highly transcribed genes following exposure to CPT in S phase, that it correlates with accumulation (pausing) of RNA Polymerase II at these positions in CPT treated cells, and that R-loops accumulate in Sae2 deficient strain (at the rDNA locus). Notably, most of these findings also apply for CtIP function in human cells: CtIP depletion triggers sensitivity to CPT that is rescued by Senataxin (the Sen1 human ortholog) overexpression and R-loops accumulate in CtIP-depleted cells, as they do in XPG-depleted cells, as previously reported. Moreover, CtIP depletion decreases ssDNA break, suggesting that, as XPG, CtIP contributes to R-loop dissolution by generating a ssDNA break promoting the activity of senataxin and other hybrids helicases. All these data identify Sae2/CtIP as a key regulator of R-loops stability on yeast and human genome. The most puzzling data came from combining both XPG and CtIP depletion, since, although their co depletion completely abolishes the generation of an ssDNA break, no R-loop accumulate in these conditions. The authors interpret these data, as a lack of R-loop extension in these conditions, that would make them undetectable.Given the threat of R-loop on genome stability that has been clearly described recently, identifying the mechanisms that function to regulate their occurrence on the genome is of prime interest. Additionally, the model presented here is very interesting and exciting. Yet I feel that (i) the model would be more convincing if the same loci were analyzed across the Figures, (see main point 1 below) and (ii) some key experiment are lacking to fully support the model (see main point 2).For the yeast part, the authors need to show that indeed, R-loops accumulates at sites that recruit sae2 upon CPT treatment. They show Figure 2E that they do accumulate on rDNA, but to be consistent with Figure 2C, and Figure 3A-B and to demonstrate convincingly the relationship at specific loci between Sae2 and R-loops, the authors need to report the accumulation of R-loops at ENO1, ADH2, FBA1, as well as SNR13, SNR5. Additionally, the authors need to extend their Sae2-flag ChIP-seq data analysis more specifically by comparing their data with DRIP-seq yeast genome profiles (see my detailed points on Figure 2).

See point #3 above, and Figure 2E. Comparison to DRIPseq data sets is discussed below.

For the mammalian cells part, the fact that codepletion of XPG and CtIP shows no R-loops accumulation but deficient ssDNA breakage is really puzzling. The authors propose that the hybrids still accumulate in these conditions but are not detectable using their assay (S9.6 and mutated cherry RNAseH). The author need to demonstrate that this is the case. Indeed, that CtIP depletion rescues R-loop accumulation in XPG-depleted cells may also indicate that CtIP lies downstream XPG mediated-ssDNA cleavage, hence challenging their model. I realize this may be difficult to assess, but this is an important piece of data that is lacking to support their current model; maybe the authors could try at one locus non denaturing sodium bisulfite treatment to detect ssDNA (as in Ginno et al., 2012 or Loomis et al., 2014)?

Much more detailed analysis is required to determine exactly what the nature of the initiating lesion is. We know there is some type of lesion, since removal of both factors results in very high sensitivity to DNA damage. We do not think the data is consistent with CtIP creating a toxic intermediate downstream of XPG action because if this were the case, CtIP depletion would rescue the sensitivity of XPG-depleted cells and this is not the case. We propose that the lesion is a nascent R-loop but it could also be simply a stalled polymerase, if secondary structure in DNA were exposed in the stalled complex. Certainly, the nature of the lesion is of interest to us, and we are working on reconstituting this in vitro, but determining this conclusively is likely the topic of an entire subsequent study.

Figure 1: Sen 1 also rescues viability upon CPT in mre 11 deficient strain. Can the author comment and integrate the MRX complex in their R-loop processing model?

We are working on that but do not have enough data to speculate on what the role of MRX(N) is in R-loop processing.

Figure 2: The ChIP-seq data should be more described, and a full supplemental Figure should be provided. Do the 45 peaks also overlap with the 176 peaks in presence of CPT? If yes, can we see some examples on browser? Can the author show a scatter plot where for each gene, the transcription level is plotted against the CPT-treated enrichment of Sae2 (normalized against gene length)? The author could also retrieve RNA Pol II mapping and perform the same analysis. Another very common way to show this type of data, would be to separate the yeast genes based on their expression level and show that Sae2 binding is statistically higher for the highly expressed genes subset compared to the low expressed set of genes.

This is what we did in Figure 2D. We look at the overlap between the sites where Sae2 is bound in S phase with CPT and compare them to the top 10% of transcribed genes (ranked by expression level) versus the same number of randomly chosen genes. The comparison with the randomly chosen genes is done 1000 times, which generates a distribution. Only in the case of S phase plus CPT is there a statistically significant difference between the overlap with the top 10% of genes and the overlap with the randomly selected genes. More detailed information about the confidence intervals for the transcription analysis is now shown in Supplementary file 1. This shows that the p value for the S+CPT immunoprecipitation overlapping with the top 10% of transcribed genes is <.0053, whereas the analysis of the S phase sample with no DNA damage yield a p value <.757. We have not done in-depth analysis of transcription levels versus Sae2 enrichment because the transcription levels we are using are from published RNA-seq data, not the S phase or S phase plus CPT cells that we are using here, which would be preferable. We are working on doing this in mammalian cells with CtIP ChIP but this is still in progress.

With respect to the comparison between the S phase peaks and the S plus CPT peaks, there are relatively few peaks that overlap. A browser view of a few of these overlapping genes is shown in Author response image 3.

**Author response image 3. respfig3:** Browser view of Sae2-ChIP at the GPR1, TBS1, and SAM2 genes in *sae2Δ* cells expressing Flag-Sae2, in S phase with CPT exposure compared to S phase with no DNA damage. Reads from the immunoprecipitated sample are shown (IP) in comparison to control immunoprecipitations performed in the absence of Flag antibody (bead control, BC).

Finally, Sae2 binding in S phase upon CPT should be compared to DRIP-seq data from the Koshland lab (Wahba et al., 2016).

9 of 45 Sae2 peaks in S phase (20%) and 36 of 176 Sae2 peaks in S phase with CPT (20%) overlap with the DRIP peaks in wild-type cells from Wahba et al., 2016. This is now stated in the text.

Given that the authors show R-loop accumulation at rDNA, can they also report the change of Sae2 recruitment upon CPT on rDNA from their Sae2 ChIP-seq? Can we also see the snapshot for PDC1, SNR13, SNR5 for Sae2 distribution in CPT treated samples?

There is substantial recruitment of Sae2 to the rDNA, which increases with CPT exposure, and we do observe Sae2 at PDC1 (Author response image 4), but there is no Sae2 detected at SNR5 or SNR13, possibly because these are non-coding RNA genes and are very short. We are not including this though as we have removed the rDNA DRIP data in favor of the more relevant RNAPII loci.

**Author response image 4. respfig4:** Browser views of Sae2-ChIP at the rDNA locus and the PDC1 locus on ch.XII in *sae2Δ* cells expressing Flag-Sae2, in S phase with CPT exposure compared to S phase with no DNA damage.

Figure 3: "We did not observe CPT induced RNA Pol II pausing at PDC1". It seems from the Figure that the author can actually see some Rbp2 increase (Figure 3B).

We corrected the description in the text.

Figure 5: In their expression study, is there some overlap between genes modified following CPT treatment and genes modified after CtIP depletion? Can the author directly show the DRIP-seq count on the different gene subsets (WT -vs +CPT, WTvsCtiP -CPT and WT vs CtIP + CPT) compared to random genes (together with p values)?

We did not do DRIP-seq. We did RNA-seq and compared the locations of genes with differences in gene expression depending on CtIP status and CPT exposure to previously published DRIPseq data (Figure 5F). This analysis showed that the genes affected by CtIP loss are overrepresented in the DRIP-seq dataset (p<0.0052 for the comparison between – /+ CtIP and the genes identified by DRIPseq). The 99% confidence intervals have been added to the supplementary file as well as cited in the text.

Figure 6: Part of the Figure 6B, can the author show the R-loop level by DRIP qPCR (only at the b-actin gene) upon XPG and CtIP depletion?

We were not able to do this with the large number of experiments requested.

Figure 8: The assay presented Figure 8D is very nice. Could the author also perform XPG only and CtIP/XPG depletion to further validate that indeed ssDNA breakage is absent in these conditions?

We would like to do this but had to prioritize other experiments based on the summary of the reviews.

Reviewer #3:In this manuscript, the role of Sae2/CTIP in R-loop accumulation and processing is explored both in yeast and mammalian cells. Motivated by understanding the unexpected sensitivity of Sae2/CTIP-deficient cells to CPT-induced 3'-ssDNA lesions, Paull and colleagues explored the role of transcriptional regulation in survival of Sae2-deficient cells. Upon finding that SEN1 expression rescues this sensitivity and that its loss enhances sensitivity, they go on to show that Sae2 is found at sites of high transcription, especially upon treatment with CPT, and that its deletion leads to transcription-dependent damage sensitivity and Pol II stalling following CPT. Similarly, they find in mammalian cells that sensitivity due to CTIP loss can be rescued by SETX expression and that RNA-DNA hybrids accumulate upon CTIP-loss or XPG-loss at laser-induced damage, based on binding of an mCherry-RNaseH sensor. Hybrids are also elevated in the absence of damage and these nucleases both at genes know to form R-loops and R-loop inducible sites. Finally, the authors show that CtIP and XPG reduce breaks formed by CPT using both comet assays and LM-PCR. Together, these studies lead the authors to propose that XPG and CtIP work in a redundant manner to process R-loops by generating a break at the R-loop site and allowing access of SETX to the hybrids for hybrid resolution.While this is an intriguing model and important function of CTIP to understand, I think that the model is somewhat premature at this point and additional results are needed to test the model and rule out other possibilities. Several experiments require additional controls as well. Of particular note, I don't think the authors have sufficiently established the proposed order of events which place CtIP and XPG upstream of SETX. Is CtIP or XPG needed for Sen1/SETX to bind the DNA at the sites of hybrid formation, tested using the RNaseH sensor and ChIP assays? This would be one prediction of the model if nuclease processing is needed to generate access for Sen1/SETX.

As discussed in point #4 above, we agree that this is an important issue and we measured SEN1 recruitment as well as SETX recruitment in yeast and human cells, respectively. We do not find support for the idea that Sae2/CtIP recruits SEN1/SETX, although we do think it is possible that Sae2/CtIP creates an intermediate that is necessary for its function. We also consider other models, as discussed in the main text.

Also, can the authors rule out that SETX and CtIP are not simply acting to reduce the same hybrid structures (albeit in different ways)? As it stands the authors need more data to substantiate their conclusions.

In the case of CtIP, we are considering the fact that all of our evidence points toward a role for the nuclease activity in promoting removal of the R-loop, and the nuclease is clearly specific for a 5ʹ flap. We have not worked with XPG in vitro but the literature shows that it also is specific for a 5ʹ flap. If these activities are working at the site of an R-loop, there is no obvious way for either enzyme independently or both together to completely remove a hybrid. So, we have postulated in our working model that SETX is acting downstream of one or both of the nucleases, since it is known to be present at sites of hybrids and to play important roles in removing hybrids. Since there is substantial evidence for R-loops forming at DSB sites (*9*) and for DSBs occurring at sites of R-loops (*10*), we are also considering the possibility that an intermediate in this pathway is a DSB (see discussion in main text).

It is also worth noting that the striking phenotype we observe with depletion of both CtIP and XPG does suggest that these are two of the major players in this process. We are not excluding the possibility that other enzymes may also be acting here though, and we consider the model to be just the best set of working hypotheses we have with the current set of data.

Where does the CPT-induced break fit into the authors' model? The break induced by CtIP/XPG would be in addition to this CPT-induced break. This is omitted from Figures and models and a somewhat confusing point.

Although we do use CPT extensively in the manuscript, the phenomenon of SEN1/SETX-induced recovery of Sae2 or CtIP-deficient cells also occurs with other forms of DNA damage including MMS and laser-induced damage. The ssDNA break formed by the Top1 adduct is not shown in the model for simplicity.

Figure 2: As a specificity control the authors should test whether Sae2 binding by ChIP is blocked by transcription inhibition. Figure 2: Since the DRIP did not work well in yeast, it would be helpful to show that Sae2 binding by ChIP is rescued by overexpression of RNaseH or Sen1 in yeast cells. Similarly, they could ask if RNaseH overexpression rescues CPT effects on survival.

We were able to work out the technical issues with DRIP in yeast and found that there are elevated R-loops at sites of high transcription and high Sae2 occupancy with DNA damage in S phase (see point # 3). We also find that the survival of *sae2* strains to CPT or MMS is promoted by overexpression of RNH1 (see Figure 1).

Figure 3: The authors need to analyze several other genes that are not expressed or expressed at lower levels to show specificity of the effect of CtIP loss at highly expressed genes. Right now they examine two genes, PDC1 and NRD1 as negative controls (if I understand this correctly) and Pol II binding is actually elevated at the first site.

We are not suggesting that Sae2 is at all highly-transcribed genes; there are many locations in the genome where transcription is high, yet we do not see Sae2 at these sites. NRD1 serves as a negative control here, where we see neither Sae2 nor accumulation of RNAPII.

Figure 4: Does RNaseH overexpression rescue viability of CTIP deletion? Also, please comment why in Figure 4 hybrids increase with DRB treatment in WT cells?

See point #2. The difference between wild-type untreated and wild-type with DRB is not statistically significant and are within the range we normally observe for wildtype cells.

Figure 4, Figure 5 and Figure 6: Is the nuclease activity of XPG required for any of the observed effects? Testing this in at least one of the assays would be helpful.

We do not have the reagents to express recombinant XPG in human cells. It is interesting that XPG was reported to associate with BRCA1, which also binds directly to CtIP, and to participate in some way in homologous recombination (*11*). We would like to know if that role and the role we are examining in this study require the nuclease activity; however, these authors reported that it is not possible to reconstitute the function of XPG with respect to homologous recombination using a transgene or overexpression so at this point we have not attempted this. It is worth trying in the future though.

Figure 5D: Does RNaseH expression in vivo and RNaseH treatment in vitro reduce the increased hybrid signal (which is actually very weak). This is an important control.

See point #1 above.

Figure 5E/F: What is the rationale for both increases and decreases in transcription overlapping with site of hybrid formation? I don't understand the implications of the RNA-seq data. Also, Figure 5F is missing in the Figure, making these data difficult to evaluate.

RNA-DNA hybrids and DNA breaks can cause both increases as well as decreases in transcription (*12*). Figure 5F does not appear to be missing.

Figure 5: What is the impact of CtIP and XPG loss, or loss of both, on the cell cycle. Are breaks or hybrid levels reduced due to alterations in cell cycle progression (e.g. failure to enter S phase) or accumulation in G2. It seems in WT cells hybrids are naturally higher in G2 and if breaks depend on S phase entry changes in cell cycle could affect the interpretation.

We have measured cell cycle progression in CtIP-depleted, XPG-depleted, and double-depleted cells and find that is not significantly different from wild-type (data not shown). We are not removing 100% of either of these factors, which may explain why we do not observe drastic changes in cell growth or cell cycle progression (Figure 4—Figure supplement 1 and Figure 4—Figure supplement 2).

Figure 6: Why are there no more breaks upon addition of CPT? Shouldn't the CPT itself lead to break formation even without the action of XPG and CtIP? Also, alkaline comet is not specific for SSB detection – these could be DSBs. I suggest the authors use caution in the way they present these data.

There are significantly more ssDNA breaks observed with CPT treatment in wild-type cells. We are showing the single-strand breaks here because we see an obvious change with DNA damage treatment and with the status of CtIP and XPG, whereas the neutral comet assay shows comparatively few breaks under the conditions used in these experiments. It is true that DSBs would also be evident in an alkaline comet assay but if this was the origin of the signal, they would be evident also in the neutral assay.

The idea that the R-loops are too small to be detected when both XPG and CtIP are lost is not well supported. Isn't it possible that there is no R-loop under these conditions and that RNAPII has simply stalled and is the toxic lesion?

Yes, this is possible. See discussion of this issue in the main text.

Can RNaseH expression rescue sensitivity due to loss of both XPG and CtIP? That is one test of this idea. Or can evidence for a smaller R-loops be obtained using bisulfite sequencing. Minimally I think other models should be considered.

We have not done RNaseH rescue of XPG+CtIP depletion. We discuss other scenarios in the main text.

[Editors' note: the author responses to the re-review follow.]

Summary:Makharashvili et al., report a novel function for Sae2/CtIP in both yeast and human cells, and propose that Sae2/CtIP, known to promote resection during DSB repair, also contributes in R-loop (DNA-RNA hybrids) dissolution, in conjunction with XPG. In this revised version, the authors have addressed to a reasonable extent my main points. More specifically they are now consistent throughout the manuscript, and report Sae2 binding and R-loop accumulation in Sae2 deleted strain at the same loci. They have also now changed their discussion to take more hypotheses into account. They have also largely controlled their data by performing RNAseH treatment to validate their results obtained with S9.6 antibody. This manuscript is a significantly improved version of the previous one. Results are now clearer and convincing. However, the key conclusion of their model that CtIP cleaves the loop needs further support and clarification before acceptance.Essential revisions:The results on which the model proposed rely on CPT-treated cells (i.e. Figure 2A-E, Figure 4A-C, Figure 5F and Figure 8) (CPT creates ssDNA breaks with Top1-cc) or cells upon laser-induced breaks (i.e. Figure 4D-H). So, the experimental conditions used implies to start with a genotoxic-induced break. It is still therefore unclear why CtIP would be required for an additional break. Indeed, authors were not able to show the proposed activity of CtIP in vitro. Authors would need to show the action of CtIP in untreated cells (no CPT, no laser irradiation), with no break, to support their model that demands that CtIP cleaves the loop. The known action of the Sae2/CtIP nuclease activity on DNA resection as a requirement to remove the RNA-DNA hybrid accumulated at a break would make much more sense and fit better with the combined effect with XPG. Therefore, either authors provide new data with untreated cells to support their model as such or alternatively include in their model the breaks caused by CPT/laser irradiation to explain the role of CtIP under such conditions.

In the current version of the manuscript, we show that CtIP depletion in untreated cells generates accumulation of R-loops (as measured by mCherry-RnaseH FACS, Figure 5A, B and by S9.6 in Figure 5C,D,E). Figure 5B specifically shows the effect of the nuclease-deficient mutant in untreated cells. Also, our quantification of ssDNA breaks at the RPL13A locus in Figure 8D is in untreated cells.

The reviewers request that existing (or exogenously induced) DNA breaks be incorporated into the model, since some of our assays utilize agents that induce breaks and under these conditions, we observe an effect of Sae2/CtIP that relates to transcription-induced DNA damage.

We have incorporated a block with an adjacent nick into a revised version of the model in Figure 7. This version also shows the nucleic acids in a manner that is closer to the most recent structures of the elongating RNA polymerase holoenzyme (Ehara et al., 2017).

References:

H. Ehara et al., Structure of the complete elongation complex of RNA polymerase II with basal factors. Science. 357, 921–924 (2017).